# Does Preprocessing Help Training Over-parameterized Neural Networks?

**Zhao Song**
Adobe Research
zsong@adobe.com

**Shuo Yang**
The University of Texas at Austin
yangshuo_ut@utexas.edu

**Ruizhe Zhang**
The University of Texas at Austin
ruizhe@utexas.edu

## Abstract

Deep neural networks have achieved impressive performance in many areas. Designing a fast and provable method for training neural networks is a fundamental question in machine learning.

The classical training method requires paying $\Omega(mnd)$ cost for both forward computation and backward computation, where $m$ is the width of the neural network, and we are given $n$ training points in $d$-dimensional space. In this paper, we propose two novel preprocessing ideas to bypass this $\Omega(mnd)$ barrier:

- First, by preprocessing the initial weights of the neural networks, we can train the neural network in $\widetilde{O}(m^{1-\Theta(1/d)}nd)$ cost per iteration.
- Second, by preprocessing the input data points, we can train the neural network in $\widetilde{O}(m^{4/5}nd)$ cost per iteration.

From the technical perspective, our result is a sophisticated combination of tools in different fields, greedy-type convergence analysis in optimization, sparsity observation in practical work, high-dimensional geometric search in data structure, concentration and anti-concentration in probability. Our results also provide theoretical insights for a large number of previously established fast training methods.

In addition, our classical algorithm can be generalized to the Quantum computation model. Interestingly, we can get a similar sublinear cost per iteration but avoid preprocessing initial weights or input data points.

## 1 Introduction

Over the last decade, deep learning has achieved dominating performance over many areas, e.g., computer vision [LBBH98, KSH12, SLJ$^+$15, HZRS16], natural language processing [CWB$^+$11, DCLT18], game playing [SHM$^+$16, SSS$^+$17] and beyond. The computational resource requirement for deep neural network training grows very quickly. Designing a fast and provable training method for neural networks is, therefore, a fundamental and demanding challenge.

Almost all deep learning models are optimized by gradient descent (or its variants). The total training time can be split into two components, the first one is the number of iterations and the second one is the cost per spent per iteration. Nearly all the iterative algorithms for acceleration can be viewed as two separate lines of research correspondingly, the first line is aiming for an algorithm that has as small as possible number of iterations, the second line is focusing on designing as efficient as possible data structures to improve the cost spent per iteration of the algorithm [Vai89, CLS19, LSZ19, JLSW20, JKL$^+$20, JSWZ21]. In this paper, our major focus is on the second line.

There are a number of practical works trying to use a nearest neighbor search data structure to speed up the per-step computation of the deep neural network training [CMF$^+$20, LXJ$^+$20, CLP$^+$21, DMZS21]. However, none of the previous work is able to give a provable guarantee. In this paper,

35th Conference on Neural Information Processing Systems (NeurIPS 2021).

our goal is to develop training algorithms that provably reduce per step time complexity. Let us consider the ReLU activation neural network and two-layer neural network[1]. Let $n$ denote the number of training data points. Let $d$ denote the dimension of each data point. Let $m$ denote the number of neurons. In each iteration of gradient descent (GD), we need to compute prediction for each point in the neural network. Each point $x_i \in \mathbb{R}^d$, requires to compute $m$ inner product in $d$ dimension. Thus, $\Omega(mnd)$ is a natural barrier for cost per iteration in training neural networks (in both forward computation and backward computation).

A natural question to ask is

> *Is it possible to improve the cost per iteration of training neural network algorithm? E.g., is $o(mnd)$ possible?*

We list our contributions as follows:

- We provide a new theoretical framework for speeding up neural network training by: 1) adopting the shifted neural tangent kernel; 2) showing that only a small fraction ($o(m)$) of neurons are activated for each input data in each training iteration; 3) identifying the sparsely activated neurons via geometric search; 4) proving that the algorithm can minimize the training loss to zero in a linear convergence rate.

- We provide two theoretical results 1) our first result (Theorem 6.1) builds a dynamic half-space report data structure for the weights of a neural network, to train neural networks in sublinear cost per iteration; 2) our second result (Theorem 6.2) builds a static half-space report data-structure for the input data points of the training data set for training a neural network in sublinear time.

**Acceleration via high-dimensional search data-structure.** High-dimensional search data structures support efficiently finding points in some geometric query regions (e.g., half-spaces, simplices, etc). Currently, there are two main approaches: one is based on Locality Sensitive Hashing (LSH) [IM98], which aims to find the close-by points (i.e., small $\ell_2$ distance [DIIM04, AR15, AIL$^+$15, ARN17, Raz17, AIR18, BIW19, DIRW20] or large inner product [SL14, SL15b, SL15a]) of a query $q \in \mathbb{R}^d$ in a given set of points $S \subset \mathbb{R}^d$. This kind of algorithms runs very fast in practice, but most of them only support approximate queries. Another approach is based on space partitioning data structures, for example, partition trees [Mat92a, Mat92b, AEM92, AC09, Cha12], $k$-$d$ trees / range trees [CT17, TOG17, Cha19], Voronoi diagrams [ADBMS98, Cha00], which can exactly search the query regions. Recent works have successfully applied high-dimensional geometric data structure to reduce the complexity of training deep learning models. SLIDE [CMF$^+$20] accelerates the forward pass by retrieving neurons with maximum inner product via an LSH-based data structure; Reformer [KKL20] similarly adopts LSH to reduce the memory usage for processing long sequence; MON-GOOSE [CLP$^+$21] accelerates the forward pass by retrieving neurons with maximum inner products via a learnable LSH-based data structure [Cha02] and lazy update framework [CLS19]. Despite the great empirical success, there is no theoretical understanding of such acceleration.

The goal of our paper is to theoretically characterize the acceleration brought by the high-dimensional geometric data structure. Specifically, our algorithm and analysis are built upon the HSR data structures [AEM92] which can find all the points that have large inner products and support efficient data update. Note that HSR comes with a stronger recovery guarantee than LSH, in the sense that HSR, whereas LSH is guaranteed to find some of those points.

**Convergence via over-parameterization.** Over the last few years, there has been a tremendous work studying the convergence result of deep neural network explicilty or implicitly based on neural tangent kernel (NTK) [JGH18], e.g. [LL18, DZPS19, AZLS19a, AZLS19b, DLL$^+$19, ADH$^+$19a, ADH$^+$19b, SY19, CGH$^+$19, ZMG19, CG19, ZG19, OS20, LSS$^+$20, JT20, ZPD$^+$20, HLSY21, BPSW21]. It has been shown that (S)GD can train a sufficiently wide NN with random initialization will converge to a small training error in polynomial steps.

---

[1]An alternative name of the two-layer neural network is "one-hidden layer neural network".

## 2 Challenges and Techniques

- Empirical works combine high-dimensional search data structures (e.g., LSH) with neural network training, however, they do not work theoretically due to the following reasons:
    - Without shifting, the number of activated (and therefore updated) neurons is $\Theta(m)$. There is no hope to theoretically prove $o(m)$ complexity (See **Challenge 1**).
    - Approximate high-dimensional search data structures might miss some important neurons, which can potentially prevent the training from converging (see **Challenge 2**).
- Our solutions are:
    - We propose a shifted ReLU activation that is guaranteed to have $o(m)$ number of activated neurons. Along with the shifted ReLU, we also propose a shifted NTK to rigorously provide a convergence guarantee (see **Solution 1**).
    - We adopt an exact high-dimensional search data structure that better couples with the shifted NTK. It takes $o(m)$ time to identify the activated neurons and fits well with the convergence analysis as it avoids missing important neurons (see **Solution 2**).

**Challenge 1: How to sparsify an over-parameterized neural network?**  To speed up the training process, we need the neural network to be "sparse", that is, for each training data $x \in \mathbb{R}^d$, the number of activated neurons is small. Then, in the forward computation, we can just evaluate a small subset of neurons. However, in the previous NTK analysis (e.g., [DZPS19]), the activation function is $\sigma(x) = \max\{\langle w_r, x \rangle, 0\}$, and the weights vectors $w_r$ are initially sampled from a standard $d$-dimensional Gaussian distribution. Then, by the symmetry of Gaussian distribution, we know that for every input data $x$, there will be about half of the neurons being activated, which means that we can only obtain a constant-factor speedup.

**Solution 1**  The problem actually comes from the activation function. In practice, people use a shifted ReLU function $\sigma_b(x) = \max\{\langle w_r, x \rangle, b_r\}$ to train neural networks. The main observation of our work is that *threshold implies sparsity*. We consider the setting where all neurons have a unified threshold parameter $b$. Then, by the concentration of Gaussian distribution, there will be $O(\exp(-b^2) \cdot m)$ activated neurons after the initialization.

The next step is to show that the number of activated neurons will not blow up too much in the following training iterations. In [DZPS19, SY19], they showed that the weights vectors are changing slowly during the training process. In our work, we open the black box of their proof and show a similar phenomenon for the shifted ReLU function. More specifically, a key component is to prove that for each training data, a large fraction of neurons will not change their status (from non-activated to activated and vice versa) in the next iteration with high probability. To achieve this, they showed that this is equivalent to the event that a standard Gaussian random variable in a small centered interval $[-R, R]$, and applied the anti-concentration inequality to upper-bound the probability. In our setting, we need to upper-bound the probability of $z \sim \mathcal{N}(0, 1)$ in a shifted interval $[b - R, b + R]$. On the one hand, we can still apply the anti-concentration inequality by showing that the probability is at most $\Pr[z \in [-R, R]]$. On the other hand, this probability is also upper-bounded by $\Pr[z > b - R]$, and for small $R$, we can apply the concentration inequality for a more accurate estimation. In the end, by some finer analysis of the probability, we can show that with high probability, the number of activated neurons in each iteration is also $O(\exp(-b^2) \cdot m)$ for each training data. If we take $b = \Theta(\sqrt{\log m})$, we only need to deal with truly sublinear in $m$ of activated neurons in the forward evaluation.

**Challenge 2: How to find the small subset of activated neurons?**  A linear scan of the neurons will lead to a time complexity linear in $m$, which we hope to avoid. Randomly sampling or using LSH for searching can potentially miss important neurons which are important for a rigorous convergence analysis.

**Solution 2**  Given the shifted ReLU function $\sigma_b(\langle w_r, x \rangle) = \max\{\langle w_r, x \rangle - b, 0\}$, the active neurons are those with weights $w_r$ lying in the half space of $\langle w_r, x \rangle - b > 0$. Finding such neurons is equivalent to a computational geometry problem: given $m$ points in $\mathbb{R}^d$, in each query and a half space $\mathcal{H}$, the goal is to output the points contained in $\mathcal{H}$. Here we use the Half-Space Reporting (HSR) data structure proposed by [AEM92]: after proper initialization, the HSR data structure can

return all points lying in the queried half space with complexity as low as $O(\log(n) + k)$, where $k$ is the number of such points. Note that the HSR data structure well couples with the shifted ReLU, as the number of activated neurons $k$ is truly sublinear in $m$ as per the setting of $b = \Theta(\sqrt{\log m})$.

# 3 Preliminaries

**Notations** For an integer $n$, we use $[n]$ to denote the set $\{1, 2, \cdots, n\}$. For a vector $x$ and $p \in \{0, 1, 2, \infty\}$, we use $\|x\|_p$ to denote the entry-wise $\ell_p$ norm of a vector. We use $I_d$ to denote $d$-dimensional identity matrix. We use $\mathcal{N}(\mu, \sigma^2)$ to denote Gaussian distribution with mean $mu$ and variance $\sigma^2$. We use $\widetilde{O}$ to hide the polylog factors.

This section is organized as follows. Section 3.1 introduces the neural network and present problem formulation. Section 3.2 presents the half-space report data-structure, Section 3.3 proposes our new sparsity-based Characterizations.

## 3.1 Problem Formulation

In this section, we introduce the neural network model we study in this work. Let us consider a two-layer ReLU activated neural network $f$ that has width $m$ and $\ell_2$ loss function. [2]

**Definition 3.1** (Prediction function and loss function). *Given* $b \in \mathbb{R}$, $x \in \mathbb{R}^d$, $W \in \mathbb{R}^{d \times m}$ *and* $a \in \mathbb{R}^m$,

$$f(W, x, a) := \frac{1}{\sqrt{m}} \sum_{r=1}^{m} a_r \sigma_b(\langle w_r, x \rangle),$$

$$L(W) := \frac{1}{2} \sum_{i=1}^{n} (f(W, x_i, a) - y_i)^2.$$

*We say function $f$ is* $2\mathsf{NN}(m, b)$ *for simplicity.*

Here $W$ are weights that connect input nodes with hidden nodes, $a_1, \cdots, a_m \in \mathbb{R}$ are the weights that connect hidden nodes with output node. The ReLU function $\sigma_b(x) := \max\{x - b, 0\}$, where $b$ is the threshold parameter. Following the literature, we mainly focus on optimizing $W \in \mathbb{R}^{d \times m}$. For weights $a \in \mathbb{R}^m$, we will never change $a$ during the training after we randomly choose them at the initialization.[3]

**Definition 3.2** (Weights at initialization). *We use the following initialization,*

- *For each $r$, we sample $w_r(0) \sim \mathcal{N}(0, I_d)$*

- *For each $r$, we sample $a_r$ from $\{-1, +1\}$ uniformly at random*

Next, we can calculate the gradient

**Fact 3.3** (Gradient of the prediction function and loss function). *For each $r \in [m]$,*

$$\frac{\partial f(W, x, a)}{\partial w_r} = \frac{1}{\sqrt{m}} a_r x \mathbf{1}_{w_r^\top x \geq b}. \tag{1}$$

*and*

$$\frac{\partial L(W)}{\partial w_r} = \frac{1}{\sqrt{m}} \sum_{i=1}^{n} (f(W, x_i, a) - y_i) a_r x_i \mathbf{1}_{\langle w_r, x_i \rangle \geq b}. \tag{2}$$

To update the weights from iteration $k$ to iteration $k + 1$, we follow the standard update rule of the GD algorithm,

$$\text{GD:} \quad W(k+1) = W(k) - \eta \cdot \Delta W(k), \quad \text{where} \quad \Delta W(k) = \frac{\partial L(W(k))}{\partial W(k)}. \tag{3}$$

---

[2]This is a very standard formulation in the literature, e.g., see [DZPS19, SY19, BPSW21]

[3]We remark, in some previous work, they do choose shift, but their shift is a random shift. In our application, it is important that the same $b$ is fixed for all neurons and never trained.

The ODE of the gradient flow is defined as

$$\frac{\mathrm{d}w_r(t)}{\mathrm{d}t} = -\frac{\partial L(W)}{\partial w_r}. \tag{4}$$

**Definition 3.4** (Error of prediction). *For each $t \in \{0, 1, \cdots, T\}$, we define $\mathrm{err}(t) \in \mathbb{R}^n$ to be the error of prediction $\mathrm{err}(t) = y - u(t)$, where $u(t) := f(W(t), a, X) \in \mathbb{R}^n$*

## 3.2 Data Structure for Half-Space Reporting

The half-space range reporting problem is an important problem in computational geometry, which is formally defined as following:

**Definition 3.5** (Half-space range reporting). *Given a set $S$ of $n$ points in $\mathbb{R}^d$. There are two operations:*

- QUERY($H$): *given a half-space $H \subset \mathbb{R}^d$, output all of the points in $S$ that contain in $H$, i.e., $S \cap H$.*

- UPDATE*: add or delete a point in $S$.*
    - INSERT($q$): *insert $q$ into $S$*
    - DELETE($q$): *delete $q$ from $S$*

*Let $\mathcal{T}_{\mathrm{init}}$ denote the pre-processing time to build the data structure, $\mathcal{T}_{\mathrm{query}}$ denote the time per query and $\mathcal{T}_{\mathrm{update}}$ time per update.*

We use the data-structure proposed in [AEM92] to solve the half-space range reporting problem, which admits the interface summarized in Algorithm 1. Intuitively, the data-structure recursively partitions the set $S$ and organizes the points in a tree data-structure. Then for a given query $(a, b)$, all $k$ points of $S$ with $\mathrm{sgn}(\langle a, x \rangle - b) \geq 0$ are reported quickly. Note that the query $(a, b)$ here defines the half-space $H$ in Definition 3.5.

---

**Algorithm 1** Half Space Report Data Structure

---

1: **data structure** HALFSPACEREPORT
2:     **procedures:**
3:         INIT($S, n, d$)         ▷ Initialize the data structure with a set $S$ of $n$ points in $\mathbb{R}^d$
4:         QUERY($a, b$)     ▷ $a, b \in \mathbb{R}^d$. Output the set $\{x \in S : \mathrm{sgn}(\langle a, x \rangle - b) \geq 0\}$
5:         ADD($x$)         ▷ Add point $x \in \mathbb{R}^d$ to $S$
6:         DELETE($x$)         ▷ Delete point $x \in \mathbb{R}^d$ from $S$
7: **end data structure**

---

Adapted from [AEM92], the algorithm comes with the following complexity:

**Corollary 3.6** ([AEM92]). *Given a set of $n$ points in $\mathbb{R}^d$, the half-space reporting problem can be solved with the following performances:*

- *Part 1. $\mathcal{T}_{\mathrm{query}}(n, d, k) = O_d(n^{1-1/\lfloor d/2 \rfloor} + k)$, amortized $\mathcal{T}_{\mathrm{update}} = O_d(\log^2(n))$.*

- *Part 2. $\mathcal{T}_{\mathrm{query}}(n, d, k) = O_d(\log(n) + k)$, amortized $\mathcal{T}_{\mathrm{update}} = O_d(n^{\lfloor d/2 \rfloor - 1})$.*

We remark that Part 1 will be used in Theorem 6.1 and Part 2 will be used in Theorem 6.2.

## 3.3 Sparsity-based Characterizations

In this section, we consider the ReLU function with a nonzero threshold: $\sigma_b(x) = \max\{0, x - b\}$, which is commonly seen in practise, and also has been considered in theoretical work [ZPD$^+$20].

We first define the set of neurons that are firing at time $t$.

**Definition 3.7** (fire set). *For each $i \in [n]$, for each $t \in \{0, 1, \cdots, T\}$, let $\mathcal{S}_{i,\mathrm{fire}}(t) \subset [m]$ denote the set of neurons that are "fire" at time $t$, i.e.,*

$$\mathcal{S}_{i,\mathrm{fire}}(t) := \{r \in [m] : \langle w_r(t), x_i \rangle > b\}.$$

*We define $k_{i,t} := |\mathcal{S}_{i,\mathrm{fire}}(t)|$, for all $t$ in $\{0, 1, \cdots, T\}$.*

We propose a new "sparsity" lemma in this work. It shows that $\sigma_b$ gives the desired sparsity.

**Lemma 3.8** (Sparsity after initialization). *Let $b > 0$ be a tunable parameter. If we use the $\sigma_b$ as the activation function, then after the initialization, with probability at least $1 - n \cdot \exp(-\Omega(m \cdot \exp(-b^2/2)))$, it holds that for each input data $x_i$, the number of activated neurons $k_{i,0}$ is at most $O(m \cdot \exp(-b^2/2))$, where $m$ is the total number of neurons.*

*Proof.* By the concentration of Gaussian distribution, the initial fire probability of a single neuron is

$$\Pr[\sigma_b(\langle w_r(0), x_i \rangle) > 0] = \Pr_{z \sim \mathcal{N}(0,1)}[z > b] \leq \exp(-b^2/2).$$

Hence, for the indicator variable $\mathbf{1}_{r \in \mathcal{S}_{i,\mathrm{fire}}(0)}$, we have

$$\mathbb{E}[\mathbf{1}_{r \in \mathcal{S}_{i,\mathrm{fire}}(0)}] \leq \exp(-b^2/2).$$

By standard concentration inequality (Lemma B.1),

$$\Pr\left[|\mathcal{S}_{i,\mathrm{fire}}(0)| > k_0 + t\right] \leq \exp\left(-\frac{t^2/2}{k_0 + t/3}\right), \forall t > 0 \tag{5}$$

where $k_0 := m \cdot \exp(-b^2/2)$. If we choose $t = k_0$, then we have:

$$\Pr\left[|\mathcal{S}_{i,\mathrm{fire}}(0)| > 2k_0\right] \leq \exp\left(-3k_0/8\right)$$

Then, by union bound over all $i \in [n]$, we have that with high probability

$$1 - n \cdot \exp(-\Omega(m \cdot \exp(-b^2/2))),$$

the number of initial fire neurons for the sample $x_i$ is bounded by $k_{i,0} \leq 2m \cdot \exp(-b^2/2)$. $\qquad\square$

The following remark gives an example of setting the threshold $b$, and will be useful for showing the sublinear complexity in the next section.

**Remark 3.9.** *If we choose $b = \sqrt{0.4 \log m}$ then $k_0 = m^{4/5}$. For $t = m^{4/5}$, Eq. (5) implies that*

$$\Pr\left[|\mathcal{S}_{i,\mathrm{fire}}(0)| > 2m^{4/5}\right] \leq \exp\left(-\min\{mR, O(m^{4/5})\}\right).$$

## 4 Training Neural Network with Half-Space Reporting Data Structure

In this section, we present two sublinear time algorithms for training over-parameterized neural networks. The first algorithm (Section 4.1) relies on building a high-dimensional search data-structure for the weights of neural network. The second algorithm (Section 4.2) is based on building a data structure for the input data points of the training set. Both of the algorithms use the HSR to quickly identify the fired neurons to avoid unnecessary calculation. The time complexity and the sketch of the proof are provided after each of the algorithms.

### 4.1 Weights Preprocessing

We first introduce the algorithm that preprocesses the weights $w_r$ for $r \in [m]$, which is commonly used in practice [CLP+21, CMF+20, KKL20]. Recall 2NN$(m, b)$ is $f(W, x, a) := \frac{1}{\sqrt{m}} \sum_{r=1}^{m} a_r \sigma_b(\langle w_r, x \rangle)$. By constructing a HSR data-structure for $w_r$'s, we can quickly find the set of active neurons $S_{i,\mathrm{fire}}$ for each of the training sample $x_i$. See pseudo-code in Algorithm 2.

In the remaining part of this section, we focus on the time complexity analysis of Algorithm 2. The convergence proof will be given in Section 5.

**Lemma 4.1** (Running time part of Theorem 6.1). *Given $n$ data points in $d$-dimensional space. Running gradient descent algorithm (Algorithm 2) on 2NN$(m, b = \sqrt{0.4 \log m})$ (Definition 3.1) the expected cost per-iteration of the gradient descent algorithm is*

$$\widetilde{O}(m^{1-\Theta(1/d)}nd).$$

---
**Algorithm 2** Training Neural Network via building a data structure of weights of the neural network
---
1: **procedure** TRAININGWITHPREPROCESSWEIGHTS($\{(x_i, y_i)\}_{i \in [n]}$,n,m,d)  ▷ Theorem 6.1
2:     Initialize $w_r, a_r$ for $r \in [m]$ and $b$ according to Definition 3.2 and Remark 3.9
3:     HALFSPACEREPORT HSR.INIT($\{w_r(0)\}_{r \in [m]}, m, d$)  ▷ Algorithm 1
4:     **for** $t = 1 \to T$ **do**
5:         $S_{i,\text{fire}} \leftarrow$ HSR.QUERY($x_i, b$) for $i \in [n]$
6:         Forward pass for $x_i$ only on neurons in $S_{i,\text{fire}}$ for $i \in [n]$
7:         Calculate gradient for $x_i$ only on neurons in $S_{i,\text{fire}}$ for $i \in [n]$
8:         Gradient update for the neurons in $\cup_{i \in [n]} S_{i,\text{fire}}$
9:         HSR.DELETE($w_r(t)$) for $r \in \cup_{i \in [n]} S_{i,\text{fire}}$
10:        HSR.ADD($w_r(t+1)$) for $r \in \cup_{i \in [n]} S_{i,\text{fire}}$
11:    **end for**
12:    **return** Trained weights $w_r(T+1)$ for $r \in [m]$
13: **end procedure**
---

---
**Algorithm 3** Training Neural Network via building a data-structure of the input data points
---
1: **procedure** TRAININGWITHPROCESSDATA($\{(x_i, y_i)\}_{i \in [n]}$,n,m,d)  ▷ Theorem 6.2
2:     Initialize $w_r, a_r$ for $r \in [m]$ and $b$ according to Definition 3.2 and Remark 3.9
3:     HALFSPACEREPORT HSR.INIT($\{x_i\}_{i \in [n]}, n, d$)  ▷ Algorithm 1
4:     $\widetilde{S}_{r,\text{fire}} \leftarrow$ HSR.QUERY($w_r(0), b$) for $r \in [m]$ ▷ $\widetilde{S}_{r,\text{fire}}$ are samples which neuron $r$ fires for
5:     $S_{i,\text{fire}} \leftarrow \{r \mid i \in \widetilde{S}_{r,\text{fire}}\}$  ▷ $S_{i,\text{fire}}$ is the set of neurons, which fire for $x_i$
6:     **for** $t = 1 \to T$ **do**
7:         Forward pass for $x_i$ only on neurons in $S_{i,\text{fire}}$ for $i \in [n]$
8:         Calculate gradient for $x_i$ only on neurons in $S_{i,\text{fire}}$ for $i \in [n]$
9:         Gradient update for the neurons in $\cup_{i \in [n]} S_{i,\text{fire}}$
10:        **for** $r \in \cup_{i \in [n]} \mathcal{S}_{i,\text{fire}}$ **do**
11:            $S_{i,\text{fire}}$.DEL($r$) for $i \in \widetilde{S}_{r,\text{fire}}$
12:            $\widetilde{S}_{r,\text{fire}} \leftarrow$ HSR.QUERY($w_r(t+1), b$)
13:            $S_{i,\text{fire}}$.ADD($r$) for $i \in \widetilde{S}_{r,\text{fire}}$
14:        **end for**
15:    **end for**
16:    **return** Trained weights $w_r(T+1)$ for $r \in [m]$
17: **end procedure**
---

*Proof.* The per-step time complexity is

$$\sum_{i=1}^{n} \mathcal{T}_{\text{QUERY}}(m, d, k_{i,t}) + (\mathcal{T}_{\text{DELETE}} + \mathcal{T}_{\text{INSERT}}) \cdot |\cup_{i \in [n]} S_{i,\text{fire}}(t)| + d \sum_{i \in [n]} k_{i,t}$$

The first term $\sum_{i=1}^{n} \mathcal{T}_{\text{QUERY}}(m, d, k_{i,t})$ corresponds to the running time of querying the active neuron set $S_{i,\text{fire}}(t)$ for all training samples $i \in [n]$. With the first result in Corollary 3.6, the complexity is bounded by $\widetilde{O}(m^{1-\Theta(1/d)}nd)$.

The second term $(\mathcal{T}_{\text{DELETE}} + \mathcal{T}_{\text{INSERT}}) \cdot |\cup_{i \in [n]} S_{i,\text{fire}}(t)|$ corresponds to updating $w_r$ in the high-dimensional search data-structure (Lines 9 and 10). Again with the first result in Corollary 3.6, we have $\mathcal{T}_{\text{DELETE}} + \mathcal{T}_{\text{INSERT}} = O(\log^2 m)$. Combining with the fact that $|\cup_{i \in [n]} S_{i,\text{fire}}(t)| \leq |\cup_{i \in [n]} S_{i,\text{fire}}(0)| \leq O(nm^{4/5})$, the second term is bounded by $O(nm^{4/5} \log^2 m)$.

The third term is the time complexity of gradient calculation restricted to the set $\mathcal{S}_{i,\text{fire}}(t)$. With the bound on $\sum_{i \in [n]} k_{i,t}$ (Lemma C.10), we have $d \sum_{i \in [n]} k_{i,t} \leq O(m^{4/5}nd)$.

Putting them together completes the proof.  □

## 4.2 Data Preprocessing

While the weights preprcessing algorithm is inspired by the common practise, the dual relationship between the input $x_i$ and model weights $w_r$ inspires us to preprocess the dataset before training (i.e., building HSR data-structure for $x_i$). This largely improves the per-iteration complexity and avoids the frequent updates of the data structure since the training data is fixed. More importantly, once the training dataset is preprocessed, it can be reused for different models or tasks, thus one does not need to perform the expensive preprocessing for each training.

The corresponding pseudocode is presented in Algorithm 3. With $x_i$ preprocessed, we can query HSR with weights $w_r$ and the result $\widetilde{S}_{r,\text{fire}}$ is the set of training samples $x_i$ for which $w_r$ fires for. Given $\widetilde{S}_{r,\text{fire}}$ for $r \in [m]$, we can easily reconstruct the set $S_{i,\text{fire}}$, which is the set of neurons fired for sample $x_i$. The forward and backward pass can then proceed similar to Algorithm 2.

At the end of each iteration, we will update $\widetilde{S}_{r,\text{fire}}$ based on the new $w_r$ estimation and update $S_{i,\text{fire}}$ accordingly. For Algorithm 3, the HSR data-structure is static for the entire training process. This is the main difference from Algorithm 2, where the HSR needs to be updated every time step to account for the changing weights $w_r$.

We defer the convergence analysis to Section 5 and focus on the time complexity analysis of Algorithm 2 in the rest of this section. We consider $d$ being a constant for the rest of this subsection.

**Lemma 4.2** (Running time part of Theorem 6.2). *Given $n$ data points in $d$-dimensional space. Running gradient descent algorithm (Algorithm 2) on* $2\mathsf{NN}(m, b = \sqrt{0.4 \log m})$ *(Definition 3.1), the expected per-iteration running time of initializing $\widetilde{S}_{r,\text{fire}}, S_{i,\text{fire}}$ for $r \in [m], i \in [n]$ is $O(m \log n + m^{4/5}n)$. The cost per iteration of the training algorithm is $O(m^{4/5}n \log n)$.*

*Proof.* We analyze the initialization and training parts separately.

**Initialization** In Lines 4 and 5, the sets $\widetilde{S}_{r,\text{fire}}, S_{i,\text{fire}}$ for $r \in [m], i \in [n]$ are initialized. For each $r \in [m]$, we need to query the data structure the set of data points $x$'s such that $\sigma_b(w_r(0)^\top x) > 0$. Hence, the running time of this step is

$$
\sum_{r=1}^{m} \mathcal{T}_{\text{query}}(n, d, \widetilde{k}_{r,0}) = O(m \log n + \sum_{r=1}^{m} \widetilde{k}_{r,0})
$$
$$
= O(m \log n + \sum_{i=1}^{n} k_{i,0})
$$
$$
= O(m \log n + m^{4/5}n).
$$

where the second step follows from $\sum_{r=1}^{m} \widetilde{k}_{r,0} = \sum_{i=1}^{n} k_{i,0}$.

**Training** Consider training the neural network for $T$ steps. For each step, first notice that the forward and backward computation parts (Line 7 - 9) are the same as previous algorithm. The time complexity is $O(m^{4/5}n \log n)$.

We next show that maintaining $\widetilde{S}_{r,\text{fire}}, r \in [m]$ and $S_{i,\text{fire}}, i \in [n]$ (Line 10 - 14) takes $O(m^{4/5}n \log n)$ time. For each fired neuron $r \in [m]$, we first remove the indices of data in the sets $S_{i,\text{fire}}$, which takes time

$$
O(1) \cdot \sum_{r \in \cup_{i \in [n]} S_{i,\text{fire}}} \widetilde{k}_{r,t} = O(1) \cdot \sum_{r=1}^{m} \widetilde{k}_{r,t} = O(m^{4/5}n).
$$

Then, we find the new set of $x$'s such that $\sigma_b(\langle w_r(t+1), x \rangle) > 0$ by querying the half-space reporting data structure. The total running time for all fired neurons is

$$
\sum_{r \in \cup_{i \in [n]} S_{i,\text{fire}}} \mathcal{T}_{\text{query}}(n, d, \widetilde{k}_{r,t+1}) \lesssim m^{4/5}n \log n + \sum_{r \in \cup_{i \in [n]} S_{i,\text{fire}}} \widetilde{k}_{r,t+1} = O(m^{4/5}n \log n)
$$

Then, we update the index sets $S_{i,\text{fire}}$ in time $O(m^{4/5}n)$. Therefore, each training step takes $O(m^{4/5}n \log n)$ time, which completes the proof. □

# 5 Convergence of Our Algorithm

We state the result of our training neural network algorithms (Lemma 5.2) can converge in certain steps. An important component in our proof is to find out a lower bound on minimum eigenvalue of the continuous Hessian matrix $\lambda_{\min}(H^{\text{cts}})$. It turns out to be an anti-concentration problem of the Gaussian random matrix. In [OS20], they gave a lower bound on $\lambda_{\min}(H^{\text{cts}})$ for ReLU function with $b = 0$, assuming the input data are separable. One of our major technical contribution is generalizing it to arbitrary $b \geq 0$.

**Proposition 5.1** (Informal version of Theorem F.1). *Given $n$ (normalized) input data points $\{x_1, x_2, \cdots, x_n\} \subseteq \mathbb{R}^d$ such that $\forall i \in [n], \|x_i\|_2 = 1$. Let parameter $\delta := \min_{i \neq j}\{\|x_i - x_j\|_2, \|x_i + x_j\|_2\}$ denote the data separability. For any shift parameter $b \geq 0$, we define shifted NTK $H^{\text{cts}} \in \mathbb{R}^{n \times n}$ as follows*

$$H^{\text{cts}}_{i,j} := \mathbb{E}_{w \sim \mathcal{N}(0, I_d)}\left[\langle x_i, x_j\rangle \cdot \mathbf{1}_{\langle w, x_i\rangle \geq b} \cdot \mathbf{1}_{\langle w, x_j\rangle \geq b}\right], \forall i \in [n], j \in [n].$$

*Then*

$$\lambda_{\min}(H^{\text{cts}}) \geq 0.01 e^{-b^2/2}\delta/n^2.$$

With proposition 5.1, we are ready to show the convergence rate of training an over-parameterized neural network with shifted ReLU function.

**Lemma 5.2** (Convergence part of Theorem 6.1 and Theorem 6.2). *Suppose input data-points are $\delta$-separable, i.e., $\delta := \min_{i \neq j}\{\|x_i - x_j\|_2, \|x_i + x_j\|_2\}$. Let $m = \text{poly}(n, 1/\delta, \log(n/\rho))$ and $\eta = O(\lambda/n^2)$. Let $b = \Theta(\sqrt{\log m})$. Then*

$$\Pr\left[\|\text{err}(k)\|_2^2 \leq (1 - \eta\lambda/2)^k \cdot \|\text{err}(0)\|_2^2, \ \forall k \in \{0, 1, \cdots, T\}\right] \geq 1 - \rho.$$

*Note that the randomness is over initialization. Eventually, we choose $T = \lambda^{-2}n^2\log(n/\epsilon)$ where $\epsilon$ is the final accuracy.*

This result shows that despite the shifted ReLU and sparsely activated neurons, we can still retain the linear convergence. Combined with the results on per-step complexity in the previous section, it gives our main theoretical results of training deep learning models with sublinear time complexity (Theorem 6.1 and Theorem 6.2).

# 6 Main Classical Results

We present two theorems (under classical computation model) of our work, showing the sublinear running time and linear convergence rate of our two algorithms. We leave the quantum application into Appendix G. The first algorithm is relying on building a high-dimensional geometric search data-structure for the weights of a neural network.

**Theorem 6.1** (Main result I, informal of Theorem E.2). *Given $n$ data points in $d$-dimensional space. We preprocess the initialization weights of the neural network. Running gradient descent algorithm (Algorithm 2) on a two-layer, $m$-width, over-parameterized ReLU neural network will minimize the training loss to zero, and the expected running time of gradient descent algorithm (per iteration) is*

$$\widetilde{O}(m^{1-\Theta(1/d)}nd).$$

The second algorithm is based on building a data structure for the input data points of the training set. Our second algorithm can further reduce the cost per iteration from $m^{1-1/d}$ to truly sublinear in $m$, e.g. $m^{4/5}$.

**Theorem 6.2** (Main result II, informal of Theorem E.2). *Given $n$ data points in $d$-dimensional space. We preprocess all the data points. Running gradient descent algorithm (Algorithm 3) on a two-layer, $m$-width, over-parameterized ReLU neural network will minimize the training loss to zero, and the expected running time of gradient descent algorithm (per iteration) is*

$$\widetilde{O}(m^{4/5}nd).$$

# 7 Discussion and Limitations

In this paper, we propose two sublinear algorithms to train neural networks. By preprocessing the weights of the neuron networks or preprocessing the training data, we rigorously prove that it is possible to train a neuron network with sublinear complexity, which overcomes the $\Omega(mnd)$ barrier in classical training methods. Our results also offer theoretical insights for many previously established fast training methods.

Our algorithm is intuitively related to the lottery tickets hypothesis [FC18]. However, our theoretical results can not be applied to explain lottery tickets immediately for two reasons: 1) the lottery ticket hypothesis focuses on pruning weights; while our results identify the important neurons. 2) the lottery ticket hypothesis identifies the weights that need to be pruned after training (by examining their magnitude), while our algorithms accelerate the training via preprocessing. It would be interesting to see how our theory can be extended to the lottery ticket hypothesis.

One limitation of our work is that the current analysis framework does not provide a convergence guarantee for combining LSH with gradient descent, which is commonly seen in many empirical works. Our proof breaks as LSH might miss important neurons which potentially ruins the convergence analysis. Instead, we refer to the HSR data structure, which provides a stronger theoretical guarantee of successfully finding all fired neurons.

## Acknowledgments and Disclosure of Funding

SY's research is supported by NSF grants 1564000 and 1934932. RZ's research is supported by NSF Grant CCF-1648712 and Scott Aaronson's Vannevar Bush Faculty Fellowship from the US Department of Defense.

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
