**Roadmap.** In Section A, we present our main algorithms. In Section B, we provide some preliminaries. In Section C, we provide sparsity analysis. We show convergence analysis in Section D. In Section E, we show how to combine the sparsity, convergence, running time all together. In Section F, we show correlation between sparsity and spectral gap of Hessian in neural tangent kernel. In Section G, we discuss how to generalize our result to quantum setting.

# A  Complete Algorithms

In this section, we present three algorithms (Alg. 4, Alg. 5 and Alg. 6) which are the complete version of Alg. 1, Alg. 2 and Alg. 3.

---

**Algorithm 4** Half Space Report Data Structure

1: **data structure** HALFSPACEREPORT
2:     **procedures:**
3:         INIT$(S, n, d)$                        ▷ Initialize the data structure with a set $S$ of $n$ points in $\mathbb{R}^d$
4:         QUERY$(a, b)$           ▷ $a, b \in \mathbb{R}^d$. Output the set $\{x \in S : \text{sgn}(\langle a, x \rangle - b) \geq 0\}$
5:         ADD$(x)$                                          ▷ Add a point $x \in \mathbb{R}^d$ to $S$
6:         DELETE$(x)$                                    ▷ Delete the point $x \in \mathbb{R}^d$ from $S$
7: **end data structure**

---

**Algorithm 5** Training Neural Network via building a data structure of weights.

1: **procedure** TRAININGWITHPREPROCESSWEIGHTS$(\{x_i\}_{i \in [n]}, \{y_i\}_{i \in [n]}, n, m, d)$        ▷ Theorem 6.1
2:     /*Initialization step*/
3:     Sample $W(0)$ and $a$ according to Definition 3.2
4:     $b \leftarrow \sqrt{0.4 \log m}$.
5:     /*A dynamic data-structure*/
6:     HALFSPACEREPORT HSR                          ▷ Algorithm 1, Part 1 of Corollary 3.6
7:     HSR.INIT$(\{w_r(0)\}_{r \in [m]}, m, d)$              ▷ It takes $\mathcal{T}_{\text{init}}(m, d)$ time
8:     /*Iterative step*/
9:     **for** $t = 0 \rightarrow T$ **do**
10:         /*Forward computation step*/
11:         **for** $i = 1 \rightarrow n$ **do**
12:             $S_{i,\text{fire}} \leftarrow$ HSR.QUERY$(x_i, b)$              ▷ It takes $\mathcal{T}_{\text{query}}(m, d, k_{i,t})$ time
13:             $u(t)_i \leftarrow \frac{1}{\sqrt{m}} \sum_{r \in \mathcal{S}_{i,\text{fire}}} a_r \cdot \sigma_b(w_r(t)^\top x_i)$        ▷ It takes $O(d \cdot k_{i,t})$ time
14:         **end for**
15:         /*Backward computation step*/
16:         $P \leftarrow 0^{n \times m}$                                          ▷ $P \in \mathbb{R}^{n \times m}$
17:         **for** $i = 1 \rightarrow n$ **do**
18:             **for** $r \in \mathcal{S}_{i,\text{fire}}$ **do**
19:                 $P_{i,r} \leftarrow \frac{1}{\sqrt{m}} a_r \cdot \sigma'_b(w_r(t)^\top x_i)$
20:             **end for**
21:         **end for**
22:         $M \leftarrow X \text{diag}(y - u(t))$                   ▷ $M \in \mathbb{R}^{d \times n}$, it takes $O(n \cdot d)$ time
23:         $\Delta W \leftarrow \underbrace{M}_{d \times n} \underbrace{P}_{n \times m}$        ▷ $\Delta W \in \mathbb{R}^{d \times m}$, it takes $O(d \cdot \text{nnz}(P))$ time, $\text{nnz}(P) = O(nm^{4/5})$
24:         $W(t+1) \leftarrow W(t) - \eta \cdot \Delta W$.
25:         /*Update data structure*/
26:         Let $Q \subset [m]$ where for each $r \in Q$, the $\Delta W_{*,r}$ is not all zeros        ▷ $|Q| \leq O(nm^{4/5})$
27:         **for** $r \in Q$ **do**
28:             HSR.DELETE$(w_r(t))$
29:             HSR.INSERT$(w_r(t+1))$
30:         **end for**
31:     **end for**
32:     **return** $W$                                          ▷ $W \in \mathbb{R}^{d \times m}$
33: **end procedure**

---

**Algorithm 6** Training Neural Network via building a data-structure of the input points.

1: **procedure** TRAININGWITHPROCESSDATA($\{x_i\}_{i\in[n]}, \{y_i\}_{i\in[n]}, n, m, d$)  ▷ Theorem 6.2
2:     /*Initialization step*/
3:     Sample $W(0)$ and $a$ according to Definition 3.2
4:     $b \leftarrow \sqrt{0.4 \log m}$.
5:     /*A static data-structure*/
6:     HALFSPACEREPORT HSR  ▷ Algorithm 1, Part 2 of Corollary 3.6
7:     HSR.INIT($\{x_i\}_{i\in[n]}, n, d$)  ▷ It takes $\mathcal{T}_{\mathsf{init}}(n, d)$ time
8:     /*Initialize $\widetilde{S}_{r,\mathrm{fire}}$ and $S_{i,\mathrm{fire}}$ */
9:                   ▷ It takes $\sum_{r=1}^m \mathcal{T}_{\mathsf{query}}(n, d, \widetilde{k}_{r,t}) = O(m \log n + m^{1/2} n)$ time
10:     $\widetilde{S}_{r,\mathrm{fire}} \leftarrow \emptyset$ for $r \in [m]$.  ▷ $\widetilde{S}_{r,\mathrm{fire}}$ is the set of samples, for which neuron $r$ fires
11:     $S_{i,\mathrm{fire}} \leftarrow \emptyset$ for $i \in [n]$.  ▷ $S_{i,\mathrm{fire}}$ is the set of neurons, which fire for $x_i$
12:     **for** $r = 1 \to m$ **do**
13:        $\widetilde{S}_{r,\mathrm{fire}} \leftarrow$ HSR.QUERY($w_r(0), b$)
14:        **for** $i \in \widetilde{S}_{r,\mathrm{fire}}$ **do**
15:           $S_{i,\mathrm{fire}}$.ADD($r$)
16:        **end for**
17:     **end for**
18:     /*Iterative step*/
19:     **for** $t = 1 \to T$ **do**
20:        /*Forward computation step*/
21:        **for** $i = 1 \to n$ **do**
22:           $u(t)_i \leftarrow \frac{1}{\sqrt{m}} \sum_{r \in \mathcal{S}_{i,\mathrm{fire}}} a_r \cdot \sigma_b(w_r(t)^\top x_i)$  ▷ It takes $O(d \cdot k_{i,t})$ time
23:        **end for**
24:        /*Backward computation step*/
25:        $P \leftarrow 0^{n \times m}$  ▷ $P \in \mathbb{R}^{n \times m}$
26:        **for** $i = 1 \to n$ **do**
27:           **for** $r \in \mathcal{S}_{i,\mathrm{fire}}$ **do**
28:              $P_{i,r} \leftarrow \frac{1}{\sqrt{m}} a_r \cdot \sigma_b'(w_r(t)^\top x_i)$
29:           **end for**
30:        **end for**
31:        $M \leftarrow X \operatorname{diag}(y - u(t))$  ▷ $M \in \mathbb{R}^{d \times n}$, it takes $O(n \cdot d)$ time
32:        $\Delta W \leftarrow \underbrace{M}_{d \times n} \underbrace{P}_{n \times m}$  ▷ $\Delta W \in \mathbb{R}^{d \times m}$, it takes $O(d \cdot \mathrm{nnz}(P))$ time, $\mathrm{nnz}(P) = O(nm^{4/5})$
33:        $W(t+1) \leftarrow W(t) - \eta \cdot \Delta W$.
34:        /*Update $\widetilde{S}_{r,\mathrm{fire}}$ and $S_{i,\mathrm{fire}}$ step*/
35:               ▷ It takes $O(\sum_{i=1}^n k_{i,t} + \sum_{r \in S_{[n],\mathrm{fire}}} \mathcal{T}_{\mathsf{query}}(n, d, \widetilde{k}_{r,t+1})) = O(n \cdot \log n \cdot m^{4/5})$
36:        $S_{[n],\mathrm{fire}} \leftarrow \cup_{i\in[n]} \mathcal{S}_{i,\mathrm{fire}}$
37:        **for** $r \in S_{[n],\mathrm{fire}}$ **do**
38:           **for** $i \in \widetilde{S}_{r,\mathrm{fire}}$ **do**  ▷ Removing old fired neuron indices. It takes $O(\widetilde{k}_{r,t})$ time
39:              $S_{i,\mathrm{fire}}$.DEL($r$)
40:           **end for**
41:           $\widetilde{S}_{r,\mathrm{fire}} \leftarrow$ HSR.QUERY($w_r(t+1), b$)  ▷ It takes $\mathcal{T}_{\mathsf{query}}(n, d, \widetilde{k}_{r,t+1})$ time
42:           **for** $i \in \widetilde{S}_{r,\mathrm{fire}}$ **do**  ▷ Adding new fired neuron indices. It takes $O(\widetilde{k}_{r,t+1})$ time
43:              $S_{i,\mathrm{fire}}$.ADD($r$)
44:           **end for**
45:        **end for**
46:     **end for**
47:     **return** $W$  ▷ $W \in \mathbb{R}^{d \times m}$
48: **end procedure**

# B Preliminaries

**Notations**   For an integer $n$, we use $[n]$ to denote the set $\{1, 2, \cdots, n\}$. For a vector $x$, we use $\|x\|_2$ to denote the entry-wise $\ell_2$ norm of a vector. We use $\mathbb{E}[]$ to denote the expectation and $\Pr[]$ to denote the probability. We use $M^\top$ to denote the transpose of $M$. We define matrix Frobenius norm as $\|M\|_F = (\sum_{i,j} M_{i,j}^2)^{1/2}$. We use $\|M\|$ to denote the operator norm of $M$. For $d \times m$ weight matrix $W$, we define $\|W\|_{\infty,2} := \max_{r\in[m]} \|w_r\|_2$. We use $x^\top y$ to denote the inner product

between vectors $x$ and $y$. We use $I_d$ to denote $d$-dimensional identity matrix. We use $\mathcal{N}(\mu, \sigma^2)$ to denote Gaussian distribution with mean $\mu$ and variance $\sigma^2$. We use $\lambda_{\min}(M)$ and $\lambda_{\max}(M)$ to denote the minimum and the maximum eigenvalue of the matrix $M$, respectively.

## B.1 Probabilities

**Lemma B.1** (Bernstein inequality [Ber24])**.** *Assume* $Z_1, \cdots, Z_n$ *are* $n$ *i.i.d. random variables.* $\forall i \in [n]$, $\mathbb{E}[Z_i] = 0$ *and* $|Z_i| \leq M$ *almost surely. Let* $Z = \sum_{i=1}^{n} Z_i$. *Then,*

$$\Pr[Z > t] \leq \exp\left(-\frac{t^2/2}{\sum_{j=1}^{n} \mathbb{E}[Z_j^2] + Mt/3}\right), \forall t > 0.$$

**Claim B.2** (Theorem 3.1 in [LS01])**.** *Let* $b > 0$ *and* $r > 0$. *Then,*

$$\exp(-b^2/2) \Pr_{x \sim \mathcal{N}(0,1)}[|x| \leq r] \leq \Pr_{x \sim \mathcal{N}(0,1)}[|x - b| \leq r] \leq \Pr_{x \sim \mathcal{N}(0,1)}[|x| \leq r].$$

**Lemma B.3** (Anti-concentration of Gaussian distribution)**.** *Let* $Z \sim \mathcal{N}(0, \sigma^2)$. *Then, for* $t > 0$,

$$\Pr[|Z| \leq t] \leq \frac{2t}{\sqrt{2\pi}\sigma}.$$

**Theorem B.4** (Theorem 5.1.1 in [Tro15])**.** *Let* $X_1, \ldots, X_m \in \mathbb{R}^{n \times n}$ *be* $m$ *independent random Hermitian matrices. Assume that* $0 \preceq X_i \preceq L \cdot I$ *for some* $L > 0$ *and for all* $i \in [m]$. *Let* $X := \sum_{i=1}^{m} X_i$. *Then, for* $\epsilon \in (0, 1]$, *we have*

$$\Pr[\lambda_{\min}(X) \leq \epsilon \lambda_{\min}(\mathbb{E}[X])] \leq n \cdot \exp(-(1 - \epsilon)^2 \lambda_{\min}(\mathbb{E}[X])/(2L)).$$

## B.2 Half-space reporting data structures

The time complexity of HSR data structure is:

**Theorem B.5** (Agarwal, Eppstein and Matousek [AEM92])**.** *Let* $d$ *be a fixed constant. Let* $t$ *be a parameter between* $n$ *and* $n^{\lfloor d/2 \rfloor}$. *There is a dynamic data structure for half-space reporting that uses* $O_{d,\epsilon}(t^{1+\epsilon})$ *space and pre-processing time,* $O_{d,\epsilon}(\frac{n}{t^{1/\lfloor d/2 \rfloor}} \log n + k)$ *time per query where* $k$ *is the output size and* $\epsilon > 0$ *is any fixed constant, and* $O_{d,\epsilon}(t^{1+\epsilon}/n)$ *amortized update time.*

As a direct corollary, we have

**Corollary B.6** (HSR data-structure time complexity [AEM92])**.** *Given a set of* $n$ *points in* $\mathbb{R}^d$, *the half-space reporting problem can be solved with the following performances:*

- *Part 1.* $\mathcal{T}_{\mathsf{init}}(n, d) = O_d(n \log n)$, $\mathcal{T}_{\mathsf{query}}(n, d, k) = O_{d,\epsilon}(n^{1-1/\lfloor d/2 \rfloor + \epsilon} + k)$, *amortized* $\mathcal{T}_{\mathsf{update}} = O_{d,\epsilon}(\log^2(n))$.

- *Part 2.* $\mathcal{T}_{\mathsf{init}}(n, d) = O_{d,\epsilon}(n^{\lfloor d/2 \rfloor + \epsilon})$, $\mathcal{T}_{\mathsf{query}}(n, d, k) = O_{d,\epsilon}(\log(n) + k)$, *amortized* $\mathcal{T}_{\mathsf{update}} = O_{d,\epsilon}(n^{\lfloor d/2 \rfloor - 1 + \epsilon})$.

## B.3 Basic algebras

**Claim B.7** ([Sch11])**.** *Let* $M_1, M_2 \in \mathbb{R}^{n \times n}$ *be two PSD matrices. Let* $M_1 \circ M_2$ *denote the Hadamard product of* $M_1$ *and* $M_2$. *Then,*

$$\lambda_{\min}(M_1 \circ M_2) \geq (\min_{i \in [n]} M_{2i,i}) \cdot \lambda_{\min}(M_1),$$

$$\lambda_{\max}(M_1 \circ M_2) \leq (\max_{i \in [n]} M_{2i,i}) \cdot \lambda_{\max}(M_1).$$

# C  Sparsity Analysis

## C.1 Bounding difference between continuous kernel and discrete kernel

In [DZPS19, SY19], they proved the following lemma for $b = 0$. Here, we provide a more general statement for any $b \geq 0$.

**Lemma C.1.** *For any shift parameter $b \geq 0$, we define continuous version of shifted NTK $H^{\mathrm{cts}}$ and discrete version of shifted NTK $H^{\mathrm{dis}}$ as:*

$$H_{i,j}^{\mathrm{cts}} := \mathop{\mathbb{E}}_{w \sim \mathcal{N}(0,I)} \left[ x_i^\top x_j \mathbf{1}_{w^\top x_i \geq b, w^\top x_j \geq b} \right],$$

$$H_{i,j}^{\mathrm{dis}} := \frac{1}{m} \sum_{r=1}^m \left[ x_i^\top x_j \mathbf{1}_{w_r^\top x_i \geq b, w_r^\top x_j \geq b} \right].$$

*We define $\lambda := \lambda_{\min}(H^{\mathrm{cts}})$.*

*Let $m = \Omega(\lambda^{-1} n \log(n/\rho))$ be number of samples of $H^{\mathrm{dis}}$, then*

$$\Pr\left[ \lambda_{\min}(H^{\mathrm{dis}}) \geq \frac{3}{4}\lambda \right] \geq 1 - \rho.$$

*Proof.* We will use the matrix Chernoff bound (Theorem B.4) to provide a lower bound on the least eigenvalue of discrete version of shifted NTK $H^{\mathrm{dis}}$.

Let $H_r := \frac{1}{m} \widetilde{X}(w_r) \widetilde{X}(w_r)^\top$, where $\widetilde{X}(w_r) \in \mathbb{R}^{d \times n}$ is defined as follows:

$$\widetilde{X}(w_r) = \begin{bmatrix} \mathbf{1}_{w_r^\top x_i \geq b} \cdot x_1 & \cdots & \mathbf{1}_{w_r^\top x_n \geq b} \cdot x_n \end{bmatrix}.$$

Hence, $H_r \succeq 0$. We need to upper-bound $\|H_r\|$. Naively, we have

$$\|H_r\| \leq \|H_r\|_F \leq \frac{n}{m},$$

since for each entry at $(i,j) \in [n] \times [n]$,

$$(H_r)_{i,j} = \frac{1}{m} x_i^\top x_j \mathbf{1}_{w_r^\top x_i \geq b, w_r^\top x_j \geq b} \leq \frac{1}{m} x_i^\top x_j \leq \frac{1}{m}.$$

Then, $H^{\mathrm{dis}} = \sum_{r=1}^m H_r$, and $\mathbb{E}[H^{\mathrm{dis}}] = H^{\mathrm{cts}}$. And we assume that $\lambda_{\min}(H^{\mathrm{cts}}) = \lambda$.

Hence, by matrix Chernoff bound (Theorem B.4) and choosing choose $m = \Omega(\lambda^{-1} n \cdot \log(n/\rho))$, we can show

$$\Pr\left[ \lambda_{\min}(H^{\mathrm{dis}}) \leq \frac{3}{4}\lambda \right] \leq n \cdot \exp(-\frac{1}{16}\lambda/(2n/m))$$

$$= n \cdot \exp(-\frac{\lambda m}{32n})$$

$$\leq \rho,$$

Thus, we finish the proof. $\qquad\square$

## C.2 Handling Hessian if perturbing weight

We present a tool which is inspired by a list of previous work [DZPS19, SY19].

**Lemma C.2** (perturbed $w$ for shifted NTK). *Let $b > 0$ and $R \leq 1/b$. Let $c > 0$ and $c' > 0$ denote two fxied constants. We define function $H$ that is mapping $\mathbb{R}^{m \times d}$ to $\mathbb{R}^{n \times n}$ as follows:*

$$\text{the } (i,j)\text{-th entry of } H(W) \text{ is } \frac{1}{m} x_i^\top x_j \sum_{r=1}^m \mathbf{1}_{w_r^\top x_i \geq b, w_r^\top x_j \geq b}.$$

*Let $\widetilde{W} \in \mathbb{R}^{d \times m}$ be $m$ vectors that are sampled from $\mathcal{N}(0, I_d)$. Consider $W \in \mathbb{R}^{d \times m}$ that satisfy, $\|\widetilde{W} - W\|_{\infty,2} \leq R$, it has*

- *Part 1, $\|H(\widetilde{W}) - H(W)\|_F \leq n \cdot \min\{c \cdot \exp(-b^2/2), 3R\}$ holds with probability at least $1 - n^2 \cdot \exp(-m \cdot \min\{c' \cdot \exp(-b^2/2), R/10\})$.*

- *Part 2, $\lambda_{\min}(H(W)) \geq \frac{3}{4}\lambda - n \cdot \min\{c \cdot \exp(-b^2/2), 3R\}$ holds with probability at least $1 - n^2 \cdot \exp(-m \cdot \min\{c' \cdot \exp(-b^2/2), R/10\}) - \rho$.*

*Proof.* Consider

$$\|H(W) - H(\widetilde{W})\|_F^2 = \sum_{i \in [n]} \sum_{j \in [n]} (H(\widetilde{W})_{i,j} - H(W)_{i,j})^2$$

$$\leq \frac{1}{m^2} \sum_{i \in [n]} \sum_{j \in [n]} \left( \sum_{r \in [m]} \mathbf{1}_{\widetilde{w}_r^\top x_i \geq b, \widetilde{w}_r^\top x_j \geq b} - \mathbf{1}_{w_r^\top x_i \geq b, w_r^\top x_j \geq b} \right)^2$$

$$= \frac{1}{m^2} \sum_{i \in [n]} \sum_{j \in [n]} \Big( \sum_{r \in [m]} s_{r,i,j,b} \Big)^2,$$

where the first step follows from definition of Frobenius norm, the last third step follows from by defining

$$s_{r,i,j,b} := \mathbf{1}_{\widetilde{w}_r^\top x_i \geq b, \widetilde{w}_r^\top x_j \geq b} - \mathbf{1}_{w_r^\top x_i \geq b, w_r^\top x_j \geq b}.$$

For simplicity, we use $s_r$ to $s_{r,i,j,b}$ (note that we fixed $(i,j)$ and $b$).

Define $A_{i,r}$ to be the event that

$$A_{i,r} = \{\exists w \in \mathbb{R}^d : \|w - w_r\|_2 \leq R, \mathbf{1}_{w^\top x_i \geq b} \neq \mathbf{1}_{w_r^\top x_i \geq b}\}.$$

Note that event $A_{i,r}$ happens iff $|w_r^\top x_i - b| \leq R$ happens.

Prior work [DZPS19, SY19] only one way to bound $\Pr[A_{i,r}]$. We present two ways of arguing the upper bound on $\Pr[A_{i,r}]$. One is anti-concentration, and the other is concentration.

By anticoncentration, (Lemma B.3),

$$\Pr[A_{i,r}] \leq \frac{2R}{\sqrt{2\pi}} \leq R.$$

By concentration,

$$\Pr[A_{i,r}] \leq \exp(-(b-R)^2/2) \leq c_1 \cdot \exp(-b^2/2).$$

where the last step follows from $R < 1/b$ and $c_1 \geq \exp(1 - R^2/2)$ is a constant.

Hence,

$$\Pr[A_{i,r}] \leq \min\{R, c_1 \exp(-b^2/2)\}.$$

If the event $\neg A_{i,r}$ happens and the event $\neg A_{j,r}$ happens, then we have

$$\left| \mathbf{1}_{\widetilde{w}_r^\top x_i \geq b, \widetilde{w}_r^\top x_j \geq b} - \mathbf{1}_{w_r^\top x_i \geq b, w_r^\top x_j \geq b} \right| = 0.$$

If the event $A_{i,r}$ happens or the event $A_{j,r}$ happens, then we obtain

$$\left| \mathbf{1}_{\widetilde{w}_r^\top x_i \geq b, \widetilde{w}_r^\top x_j \geq b} - \mathbf{1}_{w_r^\top x_i \geq b, w_r^\top x_j \geq b} \right| \leq 1.$$

**Case 1:** $c_1 \exp(-b^2/2) < R$. So we have

$$\underset{\widetilde{w}_r}{\mathbb{E}}[s_r] \leq \Pr[A_{i,r}] + \Pr[A_{j,r}]$$

$$\leq c_1 \cdot \exp(-b^2/2)$$

Now, we calculate the variance

$$\underset{\widetilde{w}_r}{\mathbb{E}}\left[ \left( s_r - \underset{\widetilde{w}_r}{\mathbb{E}}[s_r] \right)^2 \right] = \underset{\widetilde{w}_r}{\mathbb{E}}[s_r^2] - \underset{\widetilde{w}_r}{\mathbb{E}}[s_r]^2$$

$$\leq \underset{\widetilde{w}_r}{\mathbb{E}}[s_r^2]$$

$$\leq \underset{\widetilde{w}_r}{\mathbb{E}}\left[ \left( \mathbf{1}_{A_{i,r} \vee A_{j,r}} \right)^2 \right]$$

$$\leq c_1 \cdot \exp(-b^2/2).$$

Note that $|s_r| \leq 1$ for all $r$.

Define $\overline{s} = \frac{1}{m} \sum_{r=1}^{m} s_r$. Thus, we are able to use Lemma B.1,

$$\Pr\left[m \cdot \overline{s} \geq m \cdot c_1 \exp(-b^2/2) + mt\right] \leq \Pr\left[\sum_{r=1}^{m} (s_r - \mathbb{E}[s_r]) \geq mt\right]$$

$$\leq \exp\left(-\frac{m^2 t^2/2}{m \cdot c_1 \exp(-b^2/2) + mt/3}\right), \quad \forall t \geq 0.$$

Define $\overline{s} = \frac{1}{m} \sum_{r=1}^{m} s_r$. Thus, it gives

$$\Pr\left[\overline{s} \geq c_2 \cdot \exp(-b^2/2)\right] \leq \exp(-c_3 \cdot m \exp(-b^2/2)),$$

where $c_2 := 2c_1, c_3 := \frac{3}{8}c_1$ are some constants.

**Case 2:** $\exp(-b^2/2) > R$. Then, we have

$$\mathbb{E}_{\widetilde{w}_r}[s_r] \leq 2R, \quad \mathbb{E}_{\widetilde{w}_r}\left[\left(s_r - \mathbb{E}_{\widetilde{w}_r}[s_r]\right)^2\right] \leq 2R.$$

Define $\overline{s} = \frac{1}{m} \sum_{r=1}^{m} s_r$. By Lemma B.1,

$$\Pr\left[\overline{s} \geq 3R\right] \leq \exp\left(-mR/10\right).$$

**Combining two cases:**

Thus, we obtain

$$\Pr\left[\|H(\widetilde{W}) - H(W)\|_F \leq n \cdot \min\{c_2 \exp(-b^2/2), 3R\}\right]$$
$$\geq 1 - n^2 \cdot \exp(-m \cdot \min\{c_3 \exp(-b^2/2), R/10\}).$$

For the second part, by Lemma C.2, $\Pr[\lambda_{\min}(H(\widetilde{W})) \geq 0.75 \cdot \lambda] \geq 1 - \rho$. Hence,

$$\lambda_{\min}(H(W)) \geq \lambda_{\min}(H(\widetilde{W})) - \|H(\widetilde{W}) - H(W)\|$$
$$\geq \lambda_{\min}(H(\widetilde{W})) - \|H(\widetilde{W}) - H(W)\|_F$$
$$\geq 0.75 \cdot \lambda - n \cdot \min\{c_2 \cdot \exp(-b^2/2), 3R\},$$

which happens with probability $1 - n^2 \cdot \exp(-m \cdot \min\{c_3 \cdot \exp(-b^2/2), R/10\}) - \rho$ by the union bound. $\qquad \square$

## C.3 Total movement of weights

**Definition C.3** (Hessian matrix at time $t$). *For $t \geq 0$, let $H(t)$ be an $n \times n$ matrix with $(i,j)$-th entry:*

$$H(t)_{i,j} := \frac{1}{m} x_i^\top x_j \sum_{r=1}^{m} \mathbf{1}_{\langle w_r(t), x_i\rangle \geq b} \mathbf{1}_{\langle w_r(t), x_j\rangle \geq b}$$

We follow the standard notation $D_{\mathrm{cts}}$ in Lemma 3.5 in [SY19].

**Definition C.4** ($D_{\mathrm{cts}}$). *Let $y \in \mathbb{R}^n$ be the vector of the training data labels. Let $\mathrm{err}(0) \in \mathbb{R}^n$ denote the error of prediction of the neural network function (Definition 3.4). Define the actual moving distance of weight $D_{\mathrm{cts}}$ to be*

$$D_{\mathrm{cts}} := \lambda^{-1} \cdot m^{-1/2} \cdot \sqrt{n} \cdot \|\mathrm{err}(0)\|_2.$$

We state a tool from previous work [DZPS19, SY19] (more specifically, Lemma 3.4 in [DZPS19], Lemma 3.6 in [SY19]). Since adding the shift parameter $b$ to NTK doesn't affect the proof of the following lemma, thus we don't provide a proof and refer the readers to prior work.

**Lemma C.5** ([DZPS19, SY19]). *The condition $D_{\mathrm{cts}} < R$ implies $\lambda_{\min}(H(t)) \geq \lambda/2$, $\forall t \geq 0$. Let $\mathrm{err}(t)$ be defined as Definition 3.4. Further,*

1. $\|W(t) - W(0)\|_{\infty,2} \leq D_{\mathrm{cts}}$,

2. $\|\mathrm{err}(t)\|_2^2 \leq \exp(-\lambda t) \cdot \|\mathrm{err}(0)\|_2^2$.

## C.4 Bounded gradient

The proof of Lemma 3.6 in [SY19] implicitly implies the following basic property of gradient.

**Claim C.6** (Bounded gradient). *Let $\mathrm{err}(s)$ be defined as Definition 3.4. For any $0 \leq s \leq t$, We have*

$$\left\|\frac{\partial L(W(s))}{\partial w_r(s)}\right\|_2 \leq \frac{\sqrt{n}}{\sqrt{m}}\|\mathrm{err}(s)\|_2$$

$$\left\|\frac{\mathrm{d}}{\mathrm{d}s}w_r(s)\right\|_2 \leq \frac{\sqrt{n}}{\sqrt{m}}\|\mathrm{err}(s)\|_2$$

*Proof.* For the first part,

$$\left\|\frac{\partial L(W(s))}{\partial w_r(s)}\right\|_2 = \left\|\sum_{i=1}^n \mathrm{err}_i(s)\frac{1}{\sqrt{m}}a_r x_i \cdot \mathbf{1}_{w_r(s)^\top x_i \geq b}\right\|_2 \qquad \text{by Eq. (4)}$$

$$\leq \frac{1}{\sqrt{m}}\sum_{i=1}^n |\mathrm{err}_i(s)| \qquad\qquad\qquad \text{by Eq. (2)}$$

$$\leq \frac{\sqrt{n}}{\sqrt{m}}\|\mathrm{err}(s)\|_2.$$

For second part, we use ODE to prove it.

$\square$

## C.5 Upper bound on the movement of weights per iteration

The following Claim is quite standard in the literature, we omitt the details.

**Claim C.7** (Corollary 4.1 in [DZPS19], Lemma 3.8 in [SY19]). *Let $\mathrm{err}(i)$ be defined as Definition 3.4. If $\forall i \in [t], \|\mathrm{err}(i)\|_2^2 \leq (1 - \eta\lambda/2)^i \cdot \|\mathrm{err}(0)\|_2^2$, then*

$$\|W(t+1) - W_r(0)\|_{\infty,2} \leq 4\lambda^{-1}m^{-1/2} \cdot \sqrt{n} \cdot \|\mathrm{err}(0)\|_2 := D.$$

## C.6 Bounding the number of fired neuron per iteration

In this section, we will show that for $t = 0, 1, \ldots, T$, the number of fire neurons $k_{i,t} = |\mathcal{S}_{i,\mathrm{fire}}(t)|$ is small with high probability.

We define the set of neurons that are flipping at time $t$:

**Definition C.8** (flip set). *For each $i \in [n]$, for each time $t \in [T]$ let $\mathcal{S}_{i,\mathrm{flip}}(t) \subset [m]$ denote the set of neurons that are never flipped during the entire training process,*

$$\mathcal{S}_{i,\mathrm{flip}}(t) := \{r \in [m] : \ \mathrm{sgn}(\langle w_r(t), x_i\rangle - b) \neq \mathrm{sgn}(\langle w_r(t-1), x_i\rangle - b)\}.$$

Over all the iterations of training algorithm, there are some neurons that never flip states. We provide a mathematical formulation of that set,

**Definition C.9** (noflip set). *For each $i \in [n]$, let $S_i \subset [m]$ denote the set of neurons that are never flipped during the entire training process,*

$$S_i := \{r \in [m] : \forall t \in [T] \ \mathrm{sgn}(\langle w_r(t), x_i\rangle - b) = \mathrm{sgn}(\langle w_r(0), x_i\rangle - b)\}. \qquad (6)$$

In Lemma 3.8, we already show that $k_{i,0} = O(m \cdot \exp(-b^2/2))$ for all $i \in [n]$ with high probability. We can show that it also holds for $t > 0$.

**Lemma C.10** (Bounding the number of fired neuron per iteration). *Let $b \geq 0$ be a parameter, and let $\sigma_b(x) = \max\{x, b\}$ be the activation function. For each $i \in [n], t \in [T]$, $k_{i,t}$ is the number of activated neurons at the $t$-th iteration. For $0 < t \leq T$, with probability at least $1 - n \cdot \exp\left(-\Omega(m) \cdot \min\{R, \exp(-b^2/2)\}\right)$, $k_{i,t}$ is at most $O(m \exp(-b^2/2))$ for all $i \in [n]$.*

*Proof.* We prove this lemma by induction.

The base case of $t = 0$ is shown by Lemma 3.8 that $k_{i,0} = O(m \cdot \exp(-b^2/2))$ for all $i \in [n]$ with probability at least $1 - n \exp(-\Omega(m \cdot \exp(-b^2/2)))$.

Assume that the statement holds for $0, \ldots, t-1$. By Claim C.7, we know $\forall k < t$,

$$\|W(k+1) - W(0)\|_{\infty,2} < R.$$

Consider the $t$-th iteration. For each $i \in [n]$, consider the set of activated neurons $\mathcal{S}_{i,\text{fire}}$. We note that for the neurons in $S_i$, with high probability these neurons will not be activated in the $t$-th iteration if they are not activated in the $(t-1)$-th iteration. By Claim C.11, for $r \in [m]$,

$$\Pr[r \notin S_i] \leq \min\left\{R, O(\exp(-b^2/2))\right\}.$$

On the one hand, if $R < O(\exp(-b^2/2))$, then $\mathbb{E}[|\overline{S_i}|] \leq mR$. By Lemma B.1,

$$\Pr\left[|\overline{S_i}| > t\right] \leq \exp\left(-\frac{t^2/2}{mR + t/3}\right).$$

If we take $t := mR$, then we have

$$\Pr\left[|\overline{S_i}| > mR\right] \leq \exp\left(-3mR/8\right).$$

On the other hand, if $O(\exp(-b^2/2)) < R$, then $\mathbb{E}[|\overline{S_i}|] \leq O(m \exp(-b^2/2))$. By Lemma B.1, we have that

$$\Pr\left[|\overline{S_i}| > t\right] \leq \exp\left(-\frac{t^2/2}{O(m \exp(-b^2/2)) + t/3}\right).$$

If we take $t := m \exp(-b^2/2)$, we have that

$$\Pr\left[|\overline{S_i}| > m \exp(-b^2/2)\right] \leq \exp(-\Omega(m \exp(-b^2/2))).$$

Then, we know that in addition to the fire neurons in $S_{i,\text{noflip}}$, there are at most $m \cdot \min\{R, \exp(-b^2/2)\}$ neurons are activated in $t$-th iteration with high probability.

By a union bound for $i \in [n]$, we obtain with probability

$$\geq 1 - n \cdot \exp(-\Omega(m) \cdot \min\{R, \exp(-b^2/2)\}),$$

the number of activated neurons for $x_i$ at the $t$-th iteration of the algorithm is

$$k_{i,t} = |S_{i,\text{fire}}(t)| \leq k_{i,0} + m \min\{R, \exp(-b^2/2)\} \leq O(m \exp(-b^2/2)),$$

where the last step follows from $k_{i,0} = O(m \exp(-b^2/2))$ by Lemma 3.8.

The Lemma is then proved for all $t = 0, \ldots, T$. $\qquad\square$

**Claim C.11** (Bound on noflip probability). *Let $R \leq 1/b$. For $i \in [n]$, let $S_i$ be the set defined by Eq. (6).*
**Part 1.** *For $r \in [m]$, $r \notin S_i$ if and only if $|\langle w_r(0), x_i\rangle - b| < R$.*
**Part 2.** *If $w_r(0) \sim \mathcal{N}(0, I_d)$, then*

$$\Pr[r \notin S_i] \leq \min\{R, O(\exp(-b^2/2))\} \quad \forall r \in [m].$$

*Proof.* **Part 1.** We first note that $r \notin S_i \subset [m]$ is equivalent to the event that

$$\exists w \in \mathbb{R}^d, \text{s.t.} \mathbf{1}_{\langle w_r(0), x_i\rangle \geq b} \neq \mathbf{1}_{\langle w, x_i\rangle \geq b} \wedge \|w - w_r(0)\|_2 < R.$$

Assume that $\|w - w_r(0)\|_2 = R$. Then, we can write $w = w_r(0) + R \cdot v$ with $\|v\|_2 = 1$ and $\langle w, x_i\rangle = \langle w_r(0), x_i\rangle + R \cdot \langle v, x_i\rangle$.

Now, suppose there exists a $w$ such that $\mathbf{1}_{\langle w_r(0), x_i\rangle \geq b} \neq \mathbf{1}_{\langle w, x_i\rangle \geq b}$.

- If $\langle w_r(0), x_i \rangle > b$, then there exists a vector $v \in \mathbb{R}^d$ such that $R \cdot \langle v, x_i \rangle < b - \langle w_r(0), x_i \rangle$,

- If $\langle w_r(0), x_i \rangle < b$, then there exists a vector $v \in \mathbb{R}^d$ such that $R \cdot \langle v, x_i \rangle > b - \langle w_r(0), x_i \rangle$.

Since $\|x_i\|_2 = 1$ and $\langle v, x_i \rangle \in [-1, 1]$, we can see that the above conditions hold if and only if
$$b - \langle w_r(0), x_i \rangle > -R, \quad \text{and}$$
$$b - \langle w_r(0), x_i \rangle < +R.$$
In other words, $r \notin S_i$ if and only if $|\langle w_r(0), x_i \rangle - b| < R$.

**Part 2.**

We have
$$
\begin{aligned}
\Pr[r \notin S_i] &= \Pr_{z \sim \mathcal{N}(0,1)}[|z - b| < R] && \text{by } \langle w_r, x_i \rangle \sim \mathcal{N}(0,1) \\
&\leq \Pr_{z \sim \mathcal{N}(0,1)}[|z| < R] && \text{by symmetric property of Gaussian distribution} \\
&\leq \frac{2R}{\sqrt{2\pi}} && \text{by anti-concentration inequality of Gaussian (Lemma B.3)} \\
&\leq R.
\end{aligned}
$$

On the other hand, we also know
$$\Pr[r \notin S_i] \leq \Pr_{z \sim \mathcal{N}(0,1)}[z \geq b - R] \leq \exp(-(b-R)^2/2) \leq O(\exp(-b^2/2)),$$

where the last step follows from $R < 1/b$. $\qquad\qquad\square$

## D   Convergence Analysis

### D.1   Upper bound the initialization

The following Claim provides an upper bound for initialization. Prior work only shows it for $b = 0$, we generalize it to $b \geq 0$. The modification to the proof of previous Claim 3.10 in [SY19] is quite straightforward, thus we omit the details here.

**Claim D.1** (Upper bound the initialization, shited NTK version of Claim 3.10 in [SY19]). *Let $b \geq 0$ denote the NTK shifted parameter. Let parameter $\rho \in (0, 1)$ denote the failure probability. Then*
$$\Pr[\|\mathrm{err}(0)\|_2^2 = O(n(1 + b^2)\log^2(n/\rho))] \geq 1 - \rho.$$

### D.2   Bounding progress per iteration

In previous work, [SY19] define $H$ and $H^\perp$ only for $b = 0$. In this section, we generalize it to $b \geq 0$. Let us define two shifted matrices $H$ and $H^\perp$
$$H(k)_{i,j} := \frac{1}{m}\sum_{r=1}^{m}\langle x_i, x_j \rangle \mathbf{1}_{\langle w_r(k), x_i \rangle \geq b, \langle w_r(k), x_j \rangle \geq b}, \tag{7}$$

$$H(k)_{i,j}^\perp := \frac{1}{m}\sum_{r \in \overline{S}_i}\langle x_i, x_j \rangle \mathbf{1}_{\langle w_r(k), x_i \rangle \geq b, \langle w_r(k), x_j \rangle \geq b}. \tag{8}$$

We define
$$v_{1,i} := \frac{1}{\sqrt{m}}\sum_{r \in S_i} a_r(\sigma_b(w_r(k+1)^\top x_i) - \sigma_b(w_r(k)^\top x_i))$$

$$v_{2,i} := \frac{1}{\sqrt{m}}\sum_{r \in \overline{S}_i} a_r(\sigma_b(w_r(k+1)^\top x_i) - \sigma_b(w_r(k)^\top x_i)) \tag{9}$$

Following the same proof as Claim 3.9 [SY19], we can show that the following Claim. The major difference between our claim and Claim 3.9 in [SY19] is, they only proved it for the case $b = 0$. We generalize it to $b \geq 0$. The proof is several basic algebra computations, we omit the details here.

**Claim D.2** (Shifted NTK version of Claim 3.9 in [SY19])**.** *Let* $\mathrm{err}(k) = y - u(k)$ *be defined as Definition 3.4.*

$$\|\mathrm{err}(k+1)\|_2^2 = \|\mathrm{err}(k)\|_2^2 + B_1 + B_2 + B_3 + B_4,$$

*where*

$$B_1 := -2\eta \cdot \mathrm{err}(k)^\top \cdot H(k) \cdot \mathrm{err}(k),$$
$$B_2 := +2\eta \cdot \mathrm{err}(k)^\top \cdot H(k)^\perp \cdot \mathrm{err}(k),$$
$$B_3 := -2\mathrm{err}(k)^\top v_2,$$
$$B_4 := +\|u(k+1) - u(k)\|_2^2.$$

The nontrivial parts in our analysis is how to bound $B_1, B_2, B_3$ and $B_4$ for the shifted cases (We will provide a proof later). Once we can bound all these terms, we can show the following result for one iteration of the algorithm:

**Lemma D.3** (Shifted NTK version of Page 13 in [SY19])**.** *We have*

$$\|\mathrm{err}(k+1)\|_2^2 \le \|\mathrm{err}(k)\|_2^2 \cdot (1 - \eta\lambda + 4\eta n \cdot \min\{R, \exp(-b^2/2)\} + \eta^2 n^2))$$

*holds with probability at least*

$$1 - 2n^2 \cdot \exp(-\Omega(m) \cdot \min\{R, \exp(-b^2/2)\}) - \rho.$$

*Proof.* We are able to provide the following upper bound for $\|\mathrm{err}(k+1)\|_2^2$:

$$\|\mathrm{err}(k+1)\|_2^2$$
$$= \|\mathrm{err}(k)\|_2^2 + B_1 + B_2 + B_3 + B_4 \qquad\qquad\qquad \text{by Claim D.2}$$
$$\le \|\mathrm{err}(k)\|_2^2(1 - \eta\lambda + 4\eta n \cdot \min\{R, \exp(-b^2/2)\} + \eta^2 n^2) \qquad \text{by Claim D.5, D.6, D.7 and D.8}$$

$\square$

### D.3 Upper bound on the norm of dual Hessian

The proof of the following fact is similar to Fact C.1 in [SY19]. We generalize the $b = 0$ to $b \ge 0$. The same bound will hold as Fact C.1 in [SY19] if we replace $\mathbf{1}_{w_r(k)^\top x_i \ge 0}$ by $\mathbf{1}_{w_r(k)^\top x_i \ge b}$. Thus, we omit the details here.

**Fact D.4** (Shifted NTK version of Fact C.1 in [SY19])**.** *Let $b \ge 0$. Let shifted matrix $H(k)^\perp$ be defined as Eq. (8). For all $k \ge 0$, we have*

$$\|H(k)^\perp\|_F \le \frac{n}{m^2} \sum_{i=1}^n |\overline{S}_i|^2.$$

### D.4 Bounding the gradient improvement term

**Claim D.5** (Bounding the gradient improvement term)**.** *Let $H(k)$ be shifted matrix (see Eq. (7)). Assume $b \ge 0$. Denote $\rho_0 = n^2 \cdot \exp(-m \cdot \min\{c' \cdot \exp(-b^2/2), R/10\}) + \rho$. We define $B_1 := -2\eta\mathrm{err}(k)^\top H(k)\mathrm{err}(k)$. Assuming either of the following condition,*

- $R \le \frac{\lambda}{12n}$,
- $b \ge \sqrt{2 \cdot \log(4cn/\lambda)}$.

*Then, we have*

$$\Pr[B_1 \le -\eta\lambda \cdot \|\mathrm{err}(k)\|_2^2] \ge 1 - \rho_0.$$

*Proof.* By Lemma C.2, there exists constants $c, c' > 0$ such that

$$\lambda_{\min}(H(W)) \ge \frac{3}{4}\lambda - n \cdot \min\{c \cdot \exp(-b^2/2), 3R\}$$

with probability at least $1 - \rho_0$.

If we have $R \leq \frac{\lambda}{12n}$ or $b \geq \sqrt{2 \cdot \log(4cn/\lambda)}$, then

$$\lambda_{\min}(H(W)) \geq \frac{1}{2}\lambda.$$

Finally, we have

$$\mathrm{err}(k)^\top \cdot H(k) \cdot \mathrm{err}(k) \geq \|\mathrm{err}(k)\|_2^2 \cdot \lambda/2.$$

$\square$

## D.5 Bounding the blowup by the dual Hessian term

**Claim D.6** (Bounding the blowup by the dual Hessian term). *Let shifted matrix $H(k)^\perp$ be defined as Eq. (8). Let $\rho_0 = n \exp(-\Omega(m) \cdot \min\{R, \exp(-b^2/2)\})$. Let $b \geq 0$ be shifted NTK parameter. We define $B_2 := 2\eta \cdot \mathrm{err}(k)^\top \cdot H(k)^\perp \cdot \mathrm{err}(k)$. Then*

$$\Pr[B_2 \leq 2\eta n \cdot \min\{R, \exp(-b^2/2)\} \cdot \|\mathrm{err}(k)\|_2^2] \geq 1 - \rho_0.$$

*Proof.* By property of spectral norm,

$$B_2 \leq 2\eta \|\mathrm{err}(k)\|_2^2 \|H(k)^\perp\|.$$

Using Fact D.4, we have $\|H(k)^\perp\|_F \leq \frac{n}{m^2} \sum_{i=1}^n |\overline{S}_i|^2$.

By Lemma C.10, $\forall i \in \{1, 2, \cdots, n\}$, it has

$$\Pr\left[|\overline{S}_i| \leq m \cdot \min\{R, \exp(-b^2/2)\}\right] \geq 1 - \rho_0. \tag{10}$$

Hence, with probability at least $1 - \rho_0$

$$\|H(k)^\perp\|_F^2 \leq \frac{n}{m^2} \cdot n \cdot m^2 \cdot \min\{R^2, \exp(-b^2)\} = n^2 \cdot \min\{R^2, \exp(-b^2)\}.$$

Putting all together, we have

$$\|H(k)^\perp\| \leq \|H(k)^\perp\|_F \leq n \cdot \min\{R, \exp(-b^2/2)\}$$

with probability at least $1 - \rho_0$.

$\square$

## D.6 Bounding the blowup by the flip-neurons term

**Claim D.7** (Bounding the blowup by flipping neurons term). *Let $\rho_0 = n \exp(-\Omega(m) \cdot \min\{R, \exp(-b^2/2)\})$. We define $B_3 := -2\mathrm{err}(k)^\top v_2$. Let $b \geq 0$ be shifted NTK parameter. Then we have*

$$\Pr[B_3 \leq 2\eta n \cdot \min\{R, \exp(-b^2/2)\} \cdot \|\mathrm{err}(k)\|_2^2] \geq 1 - \rho_0.$$

*Proof.* Using Cauchy-Schwarz inequality, we have $B_3 \leq 2\|\mathrm{err}(k)\|_2 \cdot \|v_2\|_2$.

Then we focus on $\|v_2\|_2$,

$$
\|v_2\|_2^2 \leq \sum_{i=1}^{n} \left( \frac{\eta}{\sqrt{m}} \sum_{r \in \overline{S}_i} \left| (\frac{\partial L(W(k))}{\partial w_r(k)})^\top x_i \right| \right)^2 \qquad \text{by Eq. (9)}
$$

$$
= \frac{\eta^2}{m} \sum_{i=1}^{n} \left( \sum_{r=1}^{m} \mathbf{1}_{r \in \overline{S}_i} \left| (\frac{\partial L(W(k))}{\partial w_r(k)})^\top x_i \right| \right)^2
$$

$$
\leq \frac{\eta^2}{m} \cdot \max_{r \in [m]} \left| \frac{\partial L(W(k))}{\partial w_r(k)} \right|^2 \cdot \sum_{i=1}^{n} \left( \sum_{r=1}^{m} \mathbf{1}_{r \in \overline{S}_i} \right)^2
$$

$$
\leq \frac{\eta^2}{m} \cdot (\frac{\sqrt{n}}{\sqrt{m}} \|\mathrm{err}(k)\|_2)^2 \cdot \sum_{i=1}^{n} \left( \sum_{r=1}^{m} \mathbf{1}_{r \in \overline{S}_i} \right)^2 \qquad \text{by Claim C.6}
$$

$$
\leq \frac{\eta^2}{m} \cdot (\frac{\sqrt{n}}{\sqrt{m}} \|\mathrm{err}(k)\|_2)^2 \cdot \sum_{i=1}^{n} m^2 \cdot \min\{R^2, \exp(-b^2)\} \qquad \text{by Eq. (10)}
$$

$$
= \eta^2 n^2 \cdot \min\{R^2, \exp(-b^2)\} \cdot \|\mathrm{err}(k)\|_2^2,
$$

$\square$

## D.7 Bounding the blowup by the prediction movement term

The proof of the following Claim is quite standard and simple in literature, see Claim 3.14 in [SY19]. We omit the details here.

**Claim D.8** (Bounding the blowup by the prediction movement term)**.**

$$
B_4 \leq \eta^2 n^2 \cdot \|\mathrm{err}(k)\|_2^2.
$$

## D.8 Putting it all together

The goal of this section to combine all the convergence analysis together.

**Lemma D.9** (Convergence)**.** *Let* $\eta = \lambda/(4n^2)$, $R = \lambda/(12n)$, *let* $b \in [0, n]$, *and*

$$
m \geq \Omega(\lambda^{-4} n^4 b^2 \log^2(n/\rho)),
$$

*we have*

$$
\Pr \left[ \|\mathrm{err}(t)\|_2^2 \leq (1 - \eta\lambda/2)^t \cdot \|\mathrm{err}(0)\|_2^2 \right] \geq 1 - 2\rho.
$$

*Proof.* We know with probability $\geq 1 - 2n^2 \cdot \exp(-\Omega(m) \cdot \min\{R, \exp(-b^2/2)\}) - \rho$,

$$
\|\mathrm{err}(t+1)\|_2^2 \leq \|\mathrm{err}(t)\|_2^2 \cdot (1 - \eta\lambda + 4\eta n \cdot \min\{R, \exp(-b^2/2)\} + \eta^2 n^2)),
$$

and we want to show that

$$
1 - \eta\lambda + 4\eta n \cdot \min\{R, \exp(-b^2/2)\} + \eta^2 n^2 \leq 1 - \eta\lambda/2, \quad \text{and} \tag{11}
$$

$$
2n^2 \cdot \exp(-\Omega(m) \cdot \min\{R, \exp(-b^2/2)\}) \leq \rho. \tag{12}
$$

Claim C.7 requires the following relationship between $D$ and $R$,

$$
D = \frac{4\sqrt{n}\|\mathrm{err}(0)\|_2}{\sqrt{m}\lambda} < R
$$

By Claim D.1, we can upper bound the prediction error at the initialization,

$$
\|\mathrm{err}(0)\|_2^2 = O(nb^2 \log^2(n/\rho)),
$$

Combining the above two equations gives

$$
R > \Omega(\lambda^{-1} n m^{-1/2} b \log(n/\rho)). \tag{13}
$$

Claim D.5 (where $0 < c < e$ is a constant) requires an upper bound on $R$,[4]

$$R \leq \frac{\lambda}{12n}. \tag{14}$$

Combing the lower bound and upper bound of $R$, it implies the lower bound on $m$ in our Lemma statement.

And Lemma C.1 also requires that

$$m = \Omega(\lambda^{-1} n \log(n/\rho)). \tag{15}$$

which is dominated by the lower bound on $m$ in our lemma statement, thus we can ignore it.

Lemma C.1 and Claim C.11 require that

$$R < 1/b. \tag{16}$$

which is equivalent to

$$b < 12n/\lambda$$

However, by Theorem F.1, it will always hold for any $b > 0$.

Note that Eq. (11) can be rewritten as

$$4\eta n \cdot \min\{R, \exp(-b^2/2)\} + \eta^2 n^2 \leq \eta\lambda/2.$$

where it follows from taking $\eta := \lambda/(4n^2)$ and $R = \lambda/(12n)$.

Therefore, we can take the choice of the parameters $m, b, R$ and Eqs. (11), (12) imply

$$\Pr[\|\mathrm{err}(t+1)\|_2^2 \leq (1 - \eta\lambda/2) \cdot \|\mathrm{err}(t)\|_2^2] \geq 1 - 2\rho.$$

$\square$

## E   Combine

**Corollary E.1** (Sublinear cost per iteration). *Let $n$ denote the number of points. Let $d$ denote the dimension of points. Let $\rho \in (0, 1/10)$ denote the failure probability. Let $\delta$ be the separability of data points. For any parameter $\alpha \in (0, 1]$, we choose $b = \sqrt{0.5(1-\alpha)\log m}$, if*

$$m = \Omega((\delta^{-4} n^{10} \log^4(n/\rho))^{1/\alpha})$$

*then the sparsity is*

$$O(m^{\frac{3+\alpha}{4}}).$$

*Furthermore,*

- *If we preprocess the initial weights of the neural network, then we choose $\alpha = 1 - 1/\Theta(d)$ to get the desired running time.*

- *If we preprocess the training data points, then we choose $\alpha$ to be an arbitrarily small constant to get the desired running time.*

*Proof.* From Theorem F.1, we know

$$\lambda \geq \exp(-b^2/2) \cdot \frac{\delta}{100n^2}$$

which is equivalent to

$$\lambda^{-1} \leq \exp(b^2/2) \cdot \frac{100n^2}{\delta}.$$

---

[4]Due to the relationship between $b$ and $\lambda$, we are not allowed to choose $b$ in an arbitrary function of $\lambda$. Thus, we should only expect to use $R$ to fix the problem.

For convergence, we need

$$m = \Omega(\lambda^{-4} n^4 b^2 \log^2(n/\rho))$$

Since we know the upper bound of $\lambda^{-1}$, thus we need to choose

$$m = \Omega(\exp(4 \cdot b^2/2) \cdot \delta^{-4} \cdot n^{10} b^2 \log^2(n/\rho))$$

From sparsity, we have

$$O(m \cdot \exp(-b^2/2))$$

Let us choose $b = \sqrt{0.5(1 - \alpha) \log m}$, for any $\alpha \in (0, 1]$.

For the lower bound on $m$, we obtain

$$m \geq (\delta^{-4} n^{10} \log^4(n/\rho))^{1/\alpha}$$

For the sparsity, we obtain

$$m \cdot m^{-(1-\alpha)/4} = m^{\frac{3+\alpha}{4}}$$

$\square$

**Theorem E.2** (Main result, formal of Theorem 6.1 and 6.2). *Given $n$ data points in $d$-dimensional space. Running gradient descent algorithm on a two-layer ReLU (over-parameterized) neural network with $m$ neurons in the hidden layers is able to minimize the training loss to zero, let $\mathcal{T}_{\mathrm{init}}$ denote the preprocessing time and $\mathcal{C}_{\mathrm{iter}}$ denote the cost per iteration of gradient descent algorithm.*

- *If we preprocess the initial weights of the neural network (Algorithm 2), then*

$$\mathcal{T}_{\mathrm{init}} = O_d(m \log m), \mathcal{C}_{\mathrm{iter}} = \widetilde{O}(m^{1-\Theta(1/d)} nd).$$

- *If we preprocess the training data points (Algorithm 3), then*

$$\mathcal{T}_{\mathrm{init}} = O(n^d), \mathcal{C}_{\mathrm{iter}} = \widetilde{O}(m^{3/4+o(1)} nd).$$

## F  Bounds for the Spectral Gap with Data Separation

**Theorem F.1** (Formal version of Proposition 5.1). *Let $x_1, \ldots, x_n$ be points in $\mathbb{R}^d$ with unit Euclidean norm and $w \sim \mathcal{N}(0, I_d)$. Form the matrix $X \in \mathbb{R}^{n \times d} = [x_1 \ \ldots \ x_n]^\top$. Suppose there exists $\delta \in (0, \sqrt{2})$ such that*

$$\min_{i \neq j \in [n]} \{\|x_i - x_j\|_2, \|x_i + x_j\|_2\} \geq \delta.$$

*Let $b \geq 0$. Recall the continuous Hessian matrix $H^{\mathrm{cts}}$ is defined by*

$$H_{i,j}^{\mathrm{cts}} := \underset{w \sim \mathcal{N}(0,I)}{\mathbb{E}} \left[ x_i^\top x_j \mathbf{1}_{w^\top x_i \geq b, w^\top x_j \geq b} \right] \quad \forall (i,j) \in [n] \times [n].$$

*Let $\lambda := \lambda_{\min}(H^{\mathrm{cts}})$. Then, we have*

$$\exp(-b^2/2) \geq \lambda \geq \exp(-b^2/2) \cdot \frac{\delta}{100n^2}. \tag{17}$$

*Proof.* **Part 1: Lower bound.**

Define the covariance of the vector $\mathbf{1}_{Xw>b} \in \mathbb{R}^n$ as

$$\underset{w \sim \mathcal{N}(0,I_d)}{\mathbb{E}} \left[ (\mathbf{1}_{Xw>b})(\mathbf{1}_{Xw>b})^\top \right].$$

Then, $H^{\mathrm{cts}}$ can be written as

$$H^{\mathrm{cts}} = \underset{w \sim \mathcal{N}(0,I_d)}{\mathbb{E}} \left[ (\mathbf{1}_{Xw>b})(\mathbf{1}_{Xw>b})^\top \right] \circ XX^\top,$$

where $A \circ B$ denotes the Hadamard product between $A$ and $B$.

By Claim B.7, and since $\|x_i\|_2 = 1$ for all $i \in [n]$, we only need to show:

$$\mathop{\mathbb{E}}_{w \sim \mathcal{N}(0, I_d)} \left[ (\mathbf{1}_{Xw > b})(\mathbf{1}_{Xw > b})^\top \right] \succeq \exp(-b^2/2) \cdot \frac{\delta}{100 n^2} \cdot I_n.$$

Fix a unit length vector $a \in \mathbb{R}^n$. Suppose there exist constants $c_1, c_2$ such that

$$\Pr \left[ \left| a^\top \mathbf{1}_{Xw > b} \right| \geq c_1 \|a\|_\infty \right] \geq \frac{c_2 \delta}{n}. \tag{18}$$

This would imply that

$$
\begin{aligned}
\mathbb{E} \left[ \left( a^\top \mathbf{1}_{Xw > b} \right)^2 \right] &\geq \mathbb{E} \left[ \left| a^\top \mathbf{1}_{Xw > b} \right| \right]^2 \\
&\geq c_1^2 \|a\|_\infty^2 \left( \frac{c_2 \delta}{n} \right)^2 \\
&\geq c_1^2 c_2 \frac{\delta}{n^2},
\end{aligned}
$$

where the first step follows from Jensen's inequality, the second step follows from Markov's inequality, the last step follows from $\|a\|_2 = 1$.

Since this is true for all $a$, we find Eq. (17) with $c_1^2 c_2 = \frac{1}{100}$ by choosing $c_1 = 1/2, c_2 = 1/25$ as described later.

Hence, our goal is proving Eq. (18). Without loss of generality, assume $|a_1| = \|a\|_\infty$ and construct an orthonormal basis $Q \in \mathbb{R}^{d \times d}$ in $\mathbb{R}^d$ where the first column is equal to $x_1 \in \mathbb{R}^d$ and $Q = [x_1 \ \overline{Q}] \in \mathbb{R}^{d \times d}$. Note that $g = Q^\top w \sim \mathcal{N}(0, I_d)$ and we have

$$w = Qg = g_1 x_1 + \overline{Q}\overline{g},$$

where $g = \begin{bmatrix} g_1 \\ \overline{g} \end{bmatrix} \in \mathbb{R}^d$ and the first step follows from $QQ^\top = I_d$.

For $0 \leq \gamma \leq 1/2$, Gaussian small ball guarantees

$$\Pr[|g_1| \leq \gamma] \geq \frac{7\gamma}{10}.$$

Then, by Theorem 3.1 in [LS01] (Claim B.2), we have

$$\Pr[|g_1 - b| \leq \gamma] \geq \exp(-b^2/2) \cdot \frac{7\gamma}{10}.$$

Next, we argue that $z_i := \langle \overline{Q}\overline{g}, x_i \rangle$ is small for all $i \neq 1$. For a fixed $i \geq 2$, observe that

$$z_i \sim \mathcal{N}(0, 1 - \langle x_1, x_i \rangle^2).$$

Let $\tau_{i,1} := \langle x_i, x_1 \rangle$.

Note that $\delta$-separation implies

$$1 - |\langle x_1, x_i \rangle| = \frac{1}{2} \min\{\|x_1 - x_i\|_2^2, \|x_1 + x_i\|_2^2\} \geq \frac{\delta^2}{2}$$

Hence $|\tau_{i,1}| \leq 1 - \delta^2/2$.

Then, from Gaussian anti-concentration bound (Lemma B.3) and variance bound on $z_i$, we have

$$
\begin{aligned}
\Pr[|z_i| \leq |\tau_{i,1}|\gamma] &\leq \sqrt{\frac{2}{\pi}} \frac{|\tau_{i,1}|\gamma}{\sqrt{1 - \tau_{i,1}^2}} \\
&\leq \frac{2\gamma}{\delta\sqrt{\pi}} \\
&\leq \frac{2\gamma}{\delta},
\end{aligned}
$$

which implies that

$$\Pr[|z_i - (1 - \tau_{i,1})b| \le |\tau_{i,1}| \cdot \gamma] \le \Pr[|z_i| \le \gamma] \le \frac{2\gamma}{\delta}.$$

Hence, by union bound,

$$\Pr[\forall i \in \{2, \cdots, n\} : |z_i - (1 - \tau_{i,1})b| \le |\tau_{i,1}| \cdot \gamma] \ge 1 - n\frac{2\gamma}{\delta}$$

Define $\mathcal{E}$ to be the following event:

$$\mathcal{E} := \left\{ |g_1 - b| \le \gamma \text{ and } |z_i - (1 - \tau_{i,1})b| \le |\tau_{i,1}| \cdot \gamma, \quad \forall i \in \{2, \cdots, n\} \right\}.$$

Since $g_1 \in \mathbb{R}$ is independent of $\overline{g}$, we have

$$\Pr[\mathcal{E}] = \Pr[|g_1 - b| \le \gamma] \cdot \Pr[\forall i \in \{2, \cdots, n\} : |z_i - (1 - \tau_{i,1})b| \le |\tau_{i,1}| \cdot \gamma]$$

$$\ge \exp(-b^2/2) \cdot \frac{7\gamma}{10} \cdot (1 - 2n\gamma/\delta)$$

$$\ge \exp(-b^2/2) \cdot \frac{7\delta}{80n}.$$

where the last step follows from choosing $\gamma := \frac{\delta}{4n} \in [0, 1/2]$.

To proceed, define

$$f(g) := \langle a, \mathbf{1}_{Xw > b} \rangle$$

$$= a_1 \cdot \mathbf{1}_{g_1 > b} + \sum_{i=2}^{n} (a_i \cdot \mathbf{1}_{x_i^\top x_1 \cdot g_1 + x_i^\top \overline{Q} \overline{g} > b})$$

$$= a_1 \cdot \mathbf{1}_{g_1 > b} + \sum_{i=2}^{n} (a_i \cdot \mathbf{1}_{\tau_{i,1} \cdot g_1 + x_i^\top \overline{Q} \overline{g} > b}).$$

where the third step follows from $\tau_{i,1} = x_i^\top x_1$.

On the event $\mathcal{E}$, by Claim F.2, we have that $\mathbf{1}_{\tau_{i,1} \cdot g_1 + z_i > b} = \mathbf{1}_{z_i > (1 - \tau_{i,1})b}$.

Hence, conditioned on $\mathcal{E}$,

$$f(g) = a_1 \mathbf{1}_{g_1 > b} + \text{rest}(\overline{g}),$$

where

$$\text{rest}(\overline{g}) := \sum_{i=2}^{n} a_i \cdot \mathbf{1}_{x_i^\top \overline{Q} \overline{g} > (1 - \tau_{i,1})b}.$$

Furthermore, conditioned on $\mathcal{E}$, $g_1, \overline{g}$ are independent as $z_i$'s are function of $\overline{g}$ alone. Hence, $\mathcal{E}$ can be split into two equally likely events that are symmetric with respect to $g_1$ i.e. $g_1 \ge b$ and $g_1 < b$.

Consequently,

$$\Pr\left[ |f(g)| \ge \max\{|a_1 \mathbf{1}_{g_1 > b} + \text{rest}(\overline{g})|, |a_1 \mathbf{1}_{g_1 < b} + \text{rest}(\overline{g})|\} \,\Big|\, \mathcal{E} \right] \ge 1/2 \qquad (19)$$

Now, using $\max\{|a|, |b|\} \ge |a - b|/2$, we find

$$\Pr[|f(g)| \ge 0.5|a_1| \cdot |\mathbf{1}_{g_1 > b} - \mathbf{1}_{g_1 < b}| \,|\, \mathcal{E}]$$

$$= \Pr[|f(g)| \ge 0.5|a_1| \,|\, \mathcal{E}]$$

$$= \Pr[|f(g)| \ge 0.5\|a\|_\infty \,|\, \mathcal{E}]$$

$$\ge 1/2,$$

where $|a_1| = \|a\|_\infty$.

This yields

$$\Pr[|f(g)| \geq \|a\|_\infty/2] \geq \Pr[\mathcal{E}]/2 \geq \exp(-b^2/2) \cdot \frac{7\delta}{160n}.$$

**Part 2: Upper bound.**

$$\begin{aligned}
\lambda &= \lambda_{\min}(H^{cts}) \\
&= \min_{x \in \mathbb{R}^d: \|x\|_2 = 1} x^\top H^{cts} x \\
&\leq e_1^\top H^{cts} e_1 \\
&= (H^{cts})_{1,1} \\
&= \mathbb{E}_w\left[x_1^\top x_1 \mathbf{1}_{w^\top x_1 \geq b,}\right] \\
&= \Pr_w[w^\top x_1 \geq b] \\
&\leq \exp(-b^2/2),
\end{aligned}$$

where $e_1 := \begin{bmatrix} 1 & 0 & \cdots & 0 \end{bmatrix}^\top$, and the sixth step follows from $\|x_1\|_2 = 1$, the last step follows from the concentration of Gaussian distribution. In Line 5 and 6 of the above proof, $w$ is sampled from $\mathcal{N}(0, I_d)$. $\square$

**Claim F.2.** *Suppose $|g_1 - b| \leq \gamma$.*

- *If $\tau_{i,1} > 0$, then $|z_i - (1 - \tau_{i,1})b| > +\tau_{i,1}\gamma$ implies that $\mathbf{1}_{\tau_{i,1} \cdot g_1 + z_i > b} = \mathbf{1}_{z_i > (1-\tau_{i,1})b}$.*

- *If $\tau_{i,1} < 0$, then $|z_i - (1 - \tau_{i,1})b| > -\tau_{i,1}\gamma$ implies that $\mathbf{1}_{\tau_{i,1} \cdot g_1 + z_i > b} = \mathbf{1}_{z_i > (1-\tau_{i,1})b}$.*

*That is, if $|z_i - (1 - \tau_{i,1})b| > |\tau_{i,1}|\gamma$, then we have $\mathbf{1}_{\tau_{i,1} \cdot g_1 + z_i > b} = \mathbf{1}_{z_i > (1-\tau_{i,1})b}$.*

*Proof.* **Case 1.** We can assume $\tau_{i,1} > 0$. By assumption, we know that $g_1 \in [b - \gamma, b + \gamma]$.

Consider the forward direction first.

If $\tau_{i,1}g_1 + z_i > b$, then

$$z_i > b - \tau_{i,1}(b + \gamma) = (1 - \tau_{i,1})b - \tau_{i,1}\gamma.$$

According to the range of $z_i$, it implies $z_i > (1 - \tau_{i,1})b$.

Then, consider the backward direction.

If $z_i > (1 - \tau_{i,1})b$, then by the range of $z_i$, we have $z_i > (1 - \tau_{i,1})b + \tau_{i,1}\gamma$.

Hence,

$$\tau_{i,1}g_1 + z_i > \tau_{i,1}(b - \gamma) + (1 - \tau_{i,1})b + \tau_{i,1}\gamma = b.$$

**Case 2.** The $\tau_{i,1} < 0$ case can be proved in a similar way. $\square$

## G  Quantum Algorithm for Training Neural Network

In this section, we provide a quantum-classical hybrid approach to train neural networks with truly sub-quadratic time per iteration. The main observation is that the classical HSR data structure can be replaced with the Grover's search algorithm in quantum.

We first state our main result in below, showing the running time of our quantum training algorithm:

**Corollary G.1** (Main theorem). *Given $n$ data points in $d$-dimensional space. Running gradient descent algorithm on a two-layer, $m$-with, over-parameterized, and ReLU neural network will minimize the training loss to zero, let $\mathcal{C}_{\text{iter}}$ denote the cost per iteration of gradient descent algorithm. Then, we have*

$$\mathcal{C}_{\text{iter}} = \widetilde{O}(m^{9/10}nd).$$

*by applying Grover's search algorithm for the neurons (Algorithm 7) or the input data points (Algorithm 8).*

**Remark G.2.** *We remark that previous works ([KLP19, AHKZ20]) on training classical neural networks use the quantum linear algebra approach, which achieves quantum speedup in the linear algebra operations in the training process. For example, [KLP19] used the block encoding technique to speedup the matrix multiplication in training convolutional neural network (CNN). [AHKZ20] used the quantum inner-product estimation to reduce each neuron's computational cost. One drawback of this approach is that the quantum linear algebra computation incurs some non-negligible errors. Hence, extra efforts of error analysis are needed to guarantee that the intermediate errors will not affect the convergence of their algorithms.*

*Compared with the previous works, the only quantum component of our algorithm is Grover's search. So, we do not need to worry about the quantum algorithm's error in the training process. And we are able to use our fast training framework to exploit a sparse structure, which makes the Grover's search algorithm run very fast, and further leads to a truly sub-quadratic training algorithm.*

**Remark G.3.** *We also remark the difference between two algorithms in this quantum section the first algorithm runs Grover's search for each data point to find the activated neurons, while the second one runs Grover's search for each neuron to find the data points that make it activated. The advantage of Algorithm 8 is it uses less quantum resources, since its search space is of size $O(n)$ and the first algorithm's search space is of size $O(m)$.*

---

**Algorithm 7** Quantum-Classical Hybrid Training Neural Network, Version 1

---

1: **procedure** ORACLEPREP($i \in [n], t \in [T]$)
2:     Prepare the quantum query oracle $\mathcal{O}_{i,t}$ such that             ▷ Each oracle call takes $O(d)$ time

$$\mathcal{O}_{i,t} : |r\rangle |0\rangle \mapsto \begin{cases} |r\rangle |1\rangle & \text{if } w_r(t)^\top x_i > b, \\ |r\rangle |0\rangle & \text{otherwise.} \end{cases}$$

3: **end procedure**
4: **procedure** QTRAININGALGORITHMI($\{x_i\}_{i \in [n]}, \{y_i\}_{i \in [n]}, n, m, d$)             ▷ Corollary G.1
5:     Sample $w(0)$ and $a$ according to def. 3.2
6:     $b \leftarrow \sqrt{0.4 \log m}$.
7:     **for** $t = 0 \rightarrow T$ **do**
8:         /*Quantum part*/
9:         **for** $i = 1 \rightarrow n$ **do**
10:             $\mathcal{O}_{i,t} \leftarrow$ ORACLEPREP($i, t$)
11:             Use Grover's search with oracle $\mathcal{O}_{i,t}$ to find the set $\mathcal{S}_{i,\text{fire}} \subset [m]$
12:                             ▷ It takes $\widetilde{O}(\sqrt{m \cdot k_{i,t}} \cdot d)$ time
13:         **end for**
14:         /*Classical part*/
15:         **for** $i = 1 \rightarrow n$ **do**
16:             $u(t)_i \leftarrow \frac{1}{\sqrt{m}} \sum_{r \in \mathcal{S}_{i,\text{fire}}} a_r \sigma_b(w_r(t)^\top x_i)$             ▷ It takes $O(d \cdot k_{i,t})$ time
17:         **end for**
18:         **for** $i = 1 \rightarrow n$ **do**
19:             **for** $r \in \mathcal{S}_{i,\text{fire}}$ **do**
20:                 $P_{i,r} \leftarrow \frac{1}{\sqrt{m}} a_r \sigma'_b(w_r(t)^\top x_i)$
21:             **end for**
22:         **end for**
23:         $M \leftarrow X \operatorname{diag}(y - u(t))$             ▷ $M \in \mathbb{R}^{d \times n}$, it takes $O(n \cdot d)$ time
24:         $\Delta W \leftarrow MP$             ▷ $\Delta W \in \mathbb{R}^{d \times m}$, it takes $O(d \cdot \operatorname{nnz}(P))$ time
25:         $W(t+1) \leftarrow W(t) - \eta \cdot \Delta W$.
26:     **end for**
27:     **return** $W$
28: **end procedure**

---

We first state a famous result about the quadratic quantum speedup for the unstructured search problem using Grover's search algorithm.

**Theorem G.4** (Grover's search algorithm [Gro96, BHMT02])**.** *Given access to the evaluation oracle for an unknown function $f : [n] \rightarrow \{0, 1\}$ such that $|f^{-1}(1)| = k$ for some unknown number $k \leq n$, we can find all $i$'s in $f^{-1}(1)$ in $\widetilde{O}(\sqrt{nk})$-time quantumly.*

---

**Algorithm 8** Quantum-Classical Hybrid Training Neural Network, Version 2.

---

1: **procedure** QTRAININGALGORITHMII($\{x_i\}_{i\in[n]}$, $\{y_i\}_{i\in[n]}$,$n$,$m$,$d$)  ▷ Corollary G.1
2:    Sample $w(0)$ and $a$ according to def. 3.2
3:    $b \leftarrow \sqrt{0.4 \log m}$.
4:    /*Initialize $\widetilde{S}_{r,\mathrm{fire}}$ and $S_{i,\mathrm{fire}}$ */   ▷ It takes $\sum_{r=1}^{m} \widetilde{O}((n\widetilde{k}_{r,t})^{1/2}d) \leq O(m^{9/10}nd)$ time in total.
5:    $\widetilde{S}_{r,\mathrm{fire}} \leftarrow \emptyset$ for $r \in [m]$.
6:    $S_{i,\mathrm{fire}} \leftarrow \emptyset$ for $i \in [n]$.
7:    **for** $r = 1 \to m$ **do**
8:       $\widetilde{S}_{r,\mathrm{fire}} \leftarrow$ use Grover's serach to find all $i \in [n]$ s.t. $\sigma_b(w_r(1)^\top x_i) \neq 0$.
9:       **for** $i \in \widetilde{S}_{r,\mathrm{fire}}$ **do**
10:          $S_{i,\mathrm{fire}}.\mathrm{ADD}(r)$
11:       **end for**
12:    **end for**
13:    /*Iterative step*/
14:    **for** $t = 0 \to T$ **do**
15:       **for** $i = 1 \to n$ **do**
16:          $u(t)_i \leftarrow \frac{1}{\sqrt{m}} \sum_{r \in \mathcal{S}_{i,\mathrm{fire}}} a_r \cdot \sigma_b(w_r(t)^\top x_i)$   ▷ It takes $O(d \cdot k_{i,t})$ time
17:       **end for**
18:       $P \leftarrow 0^{n \times m}$   ▷ $P \in \mathbb{R}^{n \times m}$
19:       **for** $i = 1 \to n$ **do**
20:          **for** $r \in \mathcal{S}_{i,\mathrm{fire}}$ **do**
21:             $P_{i,r} \leftarrow \frac{1}{\sqrt{m}} a_r \cdot \sigma_b'(w_r(t)^\top x_i)$
22:          **end for**
23:       **end for**
24:       $M \leftarrow X \operatorname{diag}(y - u(t))$   ▷ $M \in \mathbb{R}^{d \times n}$, it takes $O(n \cdot d)$ time
25:       $\Delta W \leftarrow MP$   ▷ $\Delta W \in \mathbb{R}^{d \times m}$, it takes $O(m^{4/5}nd)$ time
26:       $W(t+1) \leftarrow W(t) - \eta \cdot \Delta W$.
27:       /*Update $\widetilde{S}_{r,\mathrm{fire}}$ and $S_{i,\mathrm{fire}}$ step*/   ▷ It takes $\widetilde{O}(m^{9/10}nd)$ time in total
28:       $S_{[n],\mathrm{fire}} \leftarrow \cup_{i \in [n]} \mathcal{S}_{i,\mathrm{fire}}$
29:       **for** $r \in S_{[n],\mathrm{fire}}$ **do**
30:          **for** $i \in \widetilde{S}_{r,\mathrm{fire}}$ **do**   ▷ It takes $O(\widetilde{k}_{r,t})$ time
31:             $S_{i,\mathrm{fire}}.\mathrm{DEL}(r)$
32:          **end for**
33:          $\widetilde{S}_{r,\mathrm{fire}} \leftarrow$ use Grover's search to find all $i \in [n]$ s.t. $\sigma_b(w_r(t+1)^\top x_i) \neq 0$.
34:          **for** $i \in \widetilde{S}_{r,\mathrm{fire}}$ **do**   ▷ It takes $O(\widetilde{k}_{r,t+1})$ time
35:             $S_{i,\mathrm{fire}}.\mathrm{ADD}(r)$
36:          **end for**
37:       **end for**
38:    **end for**
39:    **return** $W$   ▷ $W \in \mathbb{R}^{d \times m}$
40: **end procedure**

---

**Lemma G.5** (Running time). *For $t = 0, 1, \ldots, T$, the time complexity of the $t$-th iteration in Algorithm 7 is*

$$\widetilde{O}\Big(nd\sqrt{m} \cdot \max_{i \in [n]} \sqrt{k_{i,t}}\Big),$$

*where $k_{i,t} = |\mathcal{S}_{i,\mathrm{fire}}(t)|$.*

*Proof.* We first consider the quantum part of the algorithm, which is dominated by the for-loop at Line 9. For each $i \in [n]$, we need to find the set $\mathcal{S}_{i,\mathrm{fire}}(t)$ by Grover's search, which takes $\widetilde{O}(\sqrt{mk_{i,t}} \cdot \mathcal{T}_{\mathrm{oracle}})$ time. In our case, each oracle call takes $O(d)$ time. Hence, the quantum part's

running time is

$$\widetilde{O}\Big(\sum_{i=1}^{n}\sqrt{mk_{i,t}}\cdot d\Big)=\widetilde{O}\Big(nd\sqrt{m}\cdot\max_{i\in[n]}\sqrt{k_{i,t}}\Big).$$

Then, consider the classical part of the algorithm. Since we already get the sets $\mathcal{S}_{i,\mathrm{fire}}$, the for-loop at Line 15 takes $O(k_t d)$ time, and the for-loop at Line 18 takes $O(k_t)$ time, where $k_t=\sum_{i=1}^{n}k_{i,t}\le n\cdot\max_{i\in[n]}k_{i,t}$. Then, at Line 24, we compute the matrix product $X\operatorname{diag}(y-u(t))P$. It's easy to see that $M=X\operatorname{diag}(y-u(t))$ can be computed in time $O(nd)$. Since $P$ is a sparse matrix, $MP$ can be computed in $O(d\cdot\mathrm{nnz}(P))=O(dk_t)$ time. Namely, we maintain a data structure for all the non-zero entries of $P$. Then calculate each row of $MP$ in time $O(\mathrm{nnz}(P))$. Hence, the total running time of the classical part is $O(nd\cdot\max_{i\in[n]}k_{i,t})$.

Since $k_{i,t}\le m$ for all $i\in[n]$, the running time per iteration of Algorithm 7 is $\widetilde{O}(nd\sqrt{m}\cdot\max_{i\in[n]}\sqrt{k_{i,t}})$, which completes the proof of the lemma. $\qquad\square$

The following lemma proves the running time of Algorithm 8.

**Lemma G.6.** *For $t=0,1,\ldots,T$, the time complexity of the $t$-th iteration in Algorithm 8 is*

$$\widetilde{O}\Big(\sqrt{n}d\cdot\sum_{r=1}^{m}\sqrt{\widetilde{k}_{r,t}}\Big),$$

*where $\widetilde{k}_{r,t}=|\widetilde{\mathcal{S}}_{r,\mathrm{fire}}|$ at time $t$.*

*Proof.* For the quantum part, the difference is at Line 8, where we use Grover's search to find the data points such that the $r$-th neuron is activated. By Theorem G.4, it takes $\widetilde{O}((n\widetilde{k}_{r,0})^{1/2})$-time quantumly. And at Line 33, we re-compute $\widetilde{\mathcal{S}}_{r,\mathrm{fire}}$, which takes $\widetilde{O}((n\widetilde{k}_{r,t+1})^{1/2})$-time quantumly. Thus, the quantum running time of Algorithm 8 is $\widetilde{O}(\sum_{r\in[m]}(n\widetilde{k}_{r,t})^{1/2})$ per iteration.

The classical part is quite similar to Algorithm 6, which takes $O(nd\cdot\max_{i\in[n]}k_{i,t})$-time per iteration.

Therefore, the cost per iteration is $\widetilde{O}(\sum_{r\in[m]}(n\widetilde{k}_{r,t})^{1/2})$, and the lemma is then proved. $\qquad\square$

Combining Lemma G.5 and Lemma G.6 proves the main result of this section:

*Proof of Corollary G.1.* In Section E, we prove that $k_{i,t}=m^{4/5}$ with high probability for all $i\in[n]$ if we take $b=\sqrt{0.4\log m}$. Hence, by Lemma G.5, each iteration in Algorithm 7 takes

$$\widetilde{O}\Big(nd\sqrt{m}\cdot\max_{i\in[n]}\sqrt{k_{i,t}}\Big)=\widetilde{O}\Big(ndm^{9/10}\Big)$$

time in quantum. On the other hand, by Lemma G.6, each iteration in Algorithm 8 takes quantum time

$$\begin{aligned}\widetilde{O}\Big(\sqrt{n}d\sum_{r\in[m]}\sqrt{\widetilde{k}_{r,t}}\Big)&\le\widetilde{O}\Big(\sqrt{n}d\sqrt{m}\sum_{r\in[m]}\widetilde{k}_{r,t}\Big)\qquad\qquad\text{(Cauchy-Schwartz inequality.)}\\&=\widetilde{O}\Big(\sqrt{n}d\sqrt{m}\big(\sum_{i\in[n]}k_{i,t}\big)^{1/2}\Big)\\&=\widetilde{O}\Big(ndm^{9/10}\Big),\end{aligned}$$

where the second step is by $\sum_{r\in[m]}\widetilde{k}_{r,t}=\sum_{i\in[n]}k_{i,t}$, which completes the proof of the corollary. $\qquad\square$