# OpenReview forum: "Does Preprocessing Help Training Over-parameterized Neural Networks?"
_NeurIPS.cc/2021/Conference — NeurIPS 2021 Poster_

### Official Review · Reviewer_NBXy · 2021-07-09

**Rating:** 7
**Confidence:** 4

**Summary:**

This paper proposed two preprocessing algorithms that can reduce the computation cost per iteration on two layer ReLU neural networks. Suppose the network has width $m$, input dimension $d$, and number of training points $n$. The first proposed algorithm preprocessed the initial weights of the neural network and can reduce the cost per iteration from $\Omega(mnd)$ to $\widetilde{O}(m^{1-\Theta(1/d)}nd)$. The second algorithm preprocessed the input data points and reduce the cost per iteration to  $\widetilde{O}(m^{4/5}nd)$. The analysis is done in the NTK regime and the authors also proved linear convergence result under the preprocessing.

The algorithm fixed the bias term to a relatively large value so that given any input the number of active neurons is only $o(m).$ The algorithm only needs to do forward and backward computation on these active neurons, which saves the computation cost. In order to efficiently identity the active neurons at each iteration, the authors adopted a dynamic half-space report data structure for the weights in one algorithm and used a static half-space report data structure for the inputs in the other.


**Limitations And Societal Impact:**

The authors have adequately addressed the limitations and potential negative societal impact of their work.

**Main Review:**

Reducing the computation cost per iteration is an important topic and there are many heuristic algorithms, but there are very few theoretical guarantees for these algorithms. This paper made a good step in this direction by designing two preprocessing algorithms that can provably reduce the computation cost and also guarantees global convergence, for a two layer ReLU nets in the lazy training regime.

My main concerns for this paper is as follows:
1. Since the analysis is done in the lazy training regime and we know the NTK techniques also applies to multi-layer nets. I wonder if the results in this paper can extend to multi-layer setting. If not, it's also good to discuss what's the challenges.
2. The theory in this paper is restricted in the lazy training regime, but we know that in practice the neural network training usually does not happen in lazy training regime and the trained network outperforms a neural tangent kernel. So I am not sure if these preprocessing algorithms work in practice or not. It might be good to verify the performance of these algorithms in practice and compare with the previous preprocessing algorithms.
3. If I understand correctly, Lemma 4.1 and Lemma 4.2 holds as long as the active neurons are sparse, not necessarily in the NTK regime. The sparsity on activations is probably still true beyond the lazy training regime. So it's indeed possible to get a provable reduction on per-iteration cost beyond lazy training regime assuming the activations are sparse.

**Time Spent Reviewing:**

4

---

> ### Author Response · Authors · 2021-08-10
> **Thanks for your review!**
>
> Thanks so much for your helpful comments and great questions. Please find our answers below:
> 1.  Please see the general response.
> 2.  There are a large volume of recent works that empirically demonstrate the power of preprocessing. For instance, [MONGOOSE](https://openreview.net/forum?id=wWK7yXkULyh) accelerates the forward pass by retrieving neurons with the maximum inner product via a learnable LSH-based data-structure, which is intuitively similar to our Algorithm 2. We want to kindly emphasize that our contribution is to present a provable guarantee on efficiency and theoretically verify its convergence. Therefore, the major contribution of our work is to explain the ``weights phenomenon’’ in the training process of the over-parameterized neural networks from a theoretical perspective, and it can provide evidence for why the previous algorithms work well in practice.
> 3. We agree with the reviewer that it is possible to get a provable reduction on per-iteration cost beyond the lazy training regime. Our current analysis relies on the lazy training regime to establish a convergence guarantee, which would otherwise be technically challenging.

---

> > ### Comment · Reviewer_NBXy · 2021-08-28
> > **Thanks for the response**
> >
> > The response has addressed my concerns and I have increased my score accordingly. I think this paper made good contributions on designing provable algorithms that can reduce the computation cost. Although the results are restricted into the two-layer NTK regime, I believe many of the ideas and proof techniques in this paper are also useful in more general settings.

---

### Official Review · Reviewer_ZsjG · 2021-07-10

**Rating:** 6
**Confidence:** 4

**Summary:**

The authors propose a variant of the over-parameterized two-layer RELU network where the original RELU activation is shifted so that fewer neurons are activated. The advantage of such modification is that since fewer neurons are activated one may carefully design techniques to only update and carry out computation on the weights related to the activated neurons. In particular there exist algorithms/data structures such as HSR to efficiently search for the activated neurons. One could either build HSR data structures on the data and query the sample or vice versa, producing two variants of the algorithms. Both variants are able to improve the computational dependency on m, the number of hidden neurons, to sub-linear. Convergence guarantees are also provided for the proposed algorithm.

**Limitations And Societal Impact:**

 the authors adequately addressed the limitations and potential negative societal impact of their work

**Main Review:**

The setting of the paper is both interesting and limited.

It is interesting since usually it is considered impossible to reduce the dependency on the number of hidden neurons to sublinear, and this seems the first work on this effort. Even though the technique itself seems like a combination of several tools, I feel those techniques are combined in an organic and natural fashion so that rigorous guarantees and claims can be made.

On the other hand, the limitation of this work is also obvious. I understand this two-layer RELU setting with a fixed second layer is very popular in the NTK literature. However, the over-parameterized two-layer RELU networks are in general not working well in real applications. One particular issue is, even though the over-parameterised two layer RELU networks are able to fit the data (while the weights are only slightly changed during the training), they tend not to generalize well. This is perhaps one major reason that in applications people still tend to use networks with limited width but more layers. There seems to be more stuff that we do not understand why neural networks generalize so well beyond the two-layer over-parameterized setting.

One question regarding Lemma 4.2: Is there a dependency on d hidden in the big O notation there?


**Time Spent Reviewing:**

3 hours

---

> ### Author Response · Authors · 2021-08-10
> **Thanks for your review!**
>
> Thanks so much for taking the time to read and understand our paper, and thanks for your valuable suggestions. It is encouraging to see that you find our results and techniques interesting.
>
> For your concern about the limitations of two-layer over-parameterized neural networks, please refer to the general response.
>
> For your question, the dependence on $d$ is hidden in the big $O$ because we need to query the half-space reporting data structure to search the activated neurons. We treat $d$ as a constant for the clarity of exposition.

---

### Official Review · Reviewer_gr3M · 2021-07-16

**Rating:** 8
**Confidence:** 4

**Summary:**

This paper studied the problem of training 2-layer over-parameterized neural networks. The commonly used backpropagation algorithm has a running time of O(mnd) per iteration, which is also the running time for neural network evaluation. In this paper, the authors study if the traning could be done in o(mnd). They give affirmative answers and show two main contributions:
1. They proposed a framework for speeding up the training algorithm, where they observed that after the randomized initialization, the activated neurons are sparse. They also gave a convergence analysis for their framework.
2. They provide an instance of the framework by using half-space report data structures to select and update the weights of neurons during the training process. Their first algorithm uses a dynamic data structure to maintain the weights and achieves sublinear cost per iteration. By the dual relation between input data and neuron weights, their second algorithm uses a static data structure to store the training data and achieves a truly sublinear cost.


**Limitations And Societal Impact:**

I have following questions about this work:

1. The recent work of Lottery Ticket Hypothesis (Frankle et al., 2019) also exploits the sparsity of the neural network. How to compare this work with LTH?

2. Can the results be generalized to neural networks with more layers? Also, can the algorithms be adapted to the stochastic gradient descent setting?


    Frankle, J., & Carbin, M. The lottery ticket hypothesis: Finding sparse, trainable neural networks. ICLR 2019.

-----------------------------------

Typos

* Line 26: “the cost per spent per iteration” should remove the first “per”
* Line 46: “only a small fraction (o(m))”, it should be o(1) since the fraction is defined as (#activated neurons/#neurons)
* Line 75: “in the sense that HSR”, the sentence is incomplete
* Line 128: the definition of shifted ReLU is inconsistent with the definition in Line 104. max{<w_r,x>-b,0} \neq max{<w_r,x>,b}


**Main Review:**

Originality: The framework and methods mentioned in this paper is new. The idea of utilizing half-space report data structure to speed up the training of neural networks is novel.

Quality: This paper is of high quality. The authors provides detailed proof of their theorems.

Clarity: This paper is written clearly. The proof is organized professionally and easy to verify.

Significance: Accelerating neural networks is certainly an important topic. I would consider the following points as the strengths of this paper:
1. Their algorithms achieve sublinear cost per training iteration, which is a surprising result that breaks a long-standing barrier. And it revealed a connection between deep learning and computational geometry. It may also provide some theoretical insights for previous practical fast training algorithms.
2. They adapted the previous analysis of over-parameterized neural networks to the “shifted ReLU” setting via a nontrivial concentration and anti-concentration analysis. In particular, they used some new techniques in concentration of measure to upper bound the number of activated neurons in each iteration.
3. Authors have an interesting observation on the initialization of over-parameterized neural networks. They showed that the threshold of ReLU function can affect the number of activated neurons for each training data, making the neural network sparse.  This is not known before, and it may have more applications in deep learning theory..
4. Their speedup framework demonstrates that sparsity can help design more efficient algorithms. It may be further generalized and solve other machine learning problems.



**Time Spent Reviewing:**

5

---

> ### Author Response · Authors · 2021-08-10
> **Thanks for your review!**
>
> Thanks so much for your positive feedback! We will correct all the typos in the final version. Please find our answers below:
>
> 1. Both this work and LTH exploit the sparsity of the neural network, but from different perspectives. In our work, we show that the activated neurons for each training data are sparse in the whole training process. And we can use some geometric data structures to identify and maintain them. For LTH, they showed that the weights of the neurons are sparse vectors and this sparse sub-network can be identified by the magnitude of the weights after training, instead of by the activation right after initialization.
>
> 2. Please refer to our general response on generalizing to multi-layer over-parameterized neural networks. For SGD, we believe our technique can be combined with [(Li, Liang, 2018)](https://arxiv.org/abs/1808.01204) and other results on SGD for two-layer over-parameterized neural networks (e.g., [(Zou et. al., 2018)](https://arxiv.org/abs/1811.08888)) to speed-up the training process.  It is an excellent future direction.
>
> ```
> Li, Yuanzhi, and Yingyu Liang. "Learning overparameterized neural networks via stochastic gradient descent on structured data." arXiv preprint arXiv:1808.01204 (2018).
> Zou, Difan, et al. "Stochastic gradient descent optimizes over-parameterized deep ReLU networks." arXiv preprint arXiv:1811.08888 (2018).
> ```

---

### Author Response · Authors · 2021-08-10
**General Response**

We thank the reviewers for the comments. Here, we address a common question: can the results be generalized to multi-layer over-parameterized neural networks?

It is a very good question. As pointed out by the reviewers, in practice the two-layer over-parameterized neural network may have some limitations in fitting the data and the generalization. Hence, speeding up the training of multi-layer neural networks is an important problem. However, our result of training two-layer neural networks in *truly sublinear time* is already highly non-trivial. And we believe that our techniques are helpful for understanding and speeding up the training algorithm for multi-layer neural networks.

More specifically, we first note that the random initialization still works for multi-layer neural networks, and we can upper-bound the initially activated neurons for each training data in each layer. For the convergence, as suggested in [(Du et al., ICLR 2019)](https://arxiv.org/abs/1810.02054), the gradient flow of multi-layer neural network is
$$
\frac{du(t)}{dt} = \sum_{h=0}^H G^{(h)}(t) (y - u(t)),
$$
where $G^{(h)}(t)$ is the Gram matrix of layer $h$.  Then, as long as the matrix $\sum_{h=0}^H G^{(h)}(t)$ has a lower-bound on the min eigenvalue,  the gradient flow will converge to zero training loss at a linear convergence rate. Formal proof can be found in [(Allen-Zhu et al., 2018)](https://arxiv.org/abs/1811.03962) and [(Zou et. al., 2018)](https://arxiv.org/abs/1811.08888). We can adapt their proofs to our preprocessing settings and use a similar concentration and anti-concentration in this work to prove the convergence rate. One challenge is that for 2-layer neural networks, we can prove a lower-bound on the min eigenvalue of the Gram matrix (Proposition 5.1). But it may have some difficulty extending it to multi-layer settings because the current proof relies on some symmetries that may be lacking in multi-layer neural networks.

We thank the reviewers for suggesting this interesting future direction. We will add more discussions about generalizing our results to multi-layer settings in the final version.

We address specific questions below.

---

### Decision · Program_Chairs · 2021-09-27

**Decision:**

Accept (Poster)

**Comment:**

This paper gives new insights into pre-processing in training over-parameterized neural networks. All reviewers recommend acceptance.